# TextOCVP: Object-Centric Video Prediction with Language Guidance

**Angel Villar-Corrales**[†]                                                   *villar@ais.uni-bonn.de*
*Autonomous Intelligent Systems group,*
*Lamarr Institute for Machine Learning and Artificial Intelligence*
*and Center for Robotics, University of Bonn, Germany*

**Gjergj Plepi**[†]                                                           *plepi@ais.uni-bonn.de*
*Autonomous Intelligent Systems group,*
*Lamarr Institute for Machine Learning and Artificial Intelligence*
*and Center for Robotics, University of Bonn, Germany*

**Sven Behnke**                                                             *behnke@uni-bonn.de*
*Autonomous Intelligent Systems group,*
*Lamarr Institute for Machine Learning and Artificial Intelligence*
*and Center for Robotics, University of Bonn, Germany*

**Reviewed on OpenReview:**  *https://openreview.net/forum?id=7JEgXCyQgX*

## Abstract

Understanding and forecasting future scene states is critical for autonomous agents to plan and act effectively in complex environments. Object-centric models, with structured latent spaces, have shown promise in modeling object dynamics and interactions in order to predict future scene states, but often struggle to scale beyond simple synthetic datasets and to integrate external guidance, limiting their applicability in robotic environments. To address these limitations, we propose TextOCVP, an object-centric model for video prediction guided by textual descriptions. TextOCVP parses an observed scene into object representations, called slots, and utilizes a text-conditioned transformer predictor to forecast future object states and video frames. Our approach jointly models object dynamics and interactions while incorporating textual guidance, enabling accurate and controllable predictions. TextOCVP's structured latent space offers a more precise control of the forecasting process, outperforming several video prediction baselines on two datasets. Additionally, we show that structured object-centric representations provide superior robustness to novel scene configurations, as well as improved controllability and interpretability, enabling more precise and understandable predictions. Videos and code are available on our `project website`.

## 1 Introduction

Understanding and reasoning about the environment is essential for enabling autonomous systems to better comprehend their surroundings, predict future events, and adapt their actions accordingly. Humans achieve these capabilities by perceiving the environment as a structured composition of individual objects that interact and evolve dynamically over time (Kahneman et al., 1992). Neural networks equipped with such compositional inductive biases have shown the ability to learn structured object-centric representations of the world, which enable desirable properties, such as out-of-distribution generalization (Dittadi et al., 2022), compositionality (Greff et al., 2020), or sample efficiency (Mosbach et al., 2025).

Recent advances in unsupervised object-centric representation learning have progressed from extracting object representations in synthetic images (Locatello et al., 2020; Lin et al., 2020) to modeling objects in video (Kipf et al., 2022; Singh et al., 2022) and scaling to real-world scenes (Seitzer et al., 2023). These

---

† denotes equal contribution.

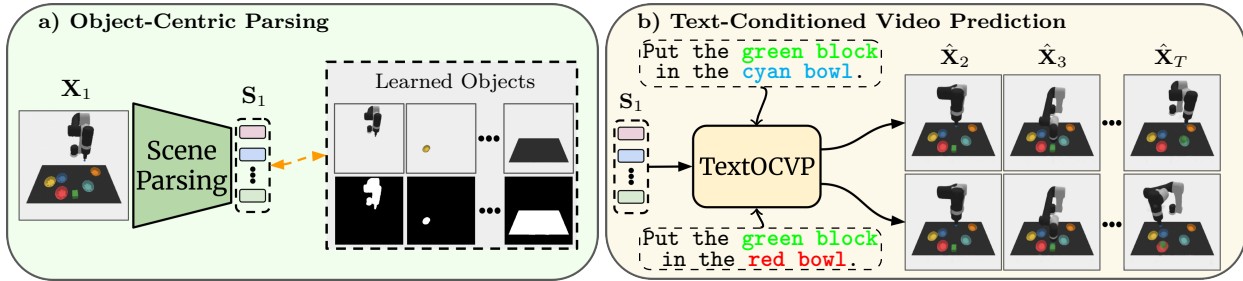

Figure 1: Overview of TextOCVP. **(a)** Our model parses a reference frame $\mathbf{X}_1$ into its object components $\mathbf{S}_1$. **(b)** Our TextOCVP predictor jointly models object dynamics and interactions guided by text, generating future object states and frames that align with the provided textual instructions.

developments have enabled object-level dynamics modeling for future prediction and planning. Notably, approaches like SlotFormer (Wu et al., 2023) or OCVP (Villar-Corrales et al., 2023) introduced object-centric prediction models that explicitly model spatio-temporal relationships between objects, shifting away from image-level approaches that ignore scene compositionality. Despite these advancements, current object-centric methods struggle with complex object appearances and dynamics, and lack mechanisms to incorporate external guidance, thus limiting their scalability and broader applicability.

To address these challenges, we propose *TextOCVP*, a novel object-centric model for video prediction guided by textual instructions, illustrated in Fig. 1. Given a reference image and text instruction, TextOCVP extracts object representations and predicts their evolution using a text-conditioned object-centric transformer. This predictor forecasts future object states by explicitly modeling their dynamics and interactions over time, while integrating textual information via a text-to-slot attention mechanism. By jointly modeling spatio-temporal object relationships and incorporating textual guidance, TextOCVP predicts future object states and frames aligned with the input instruction.

We evaluate our approach for text-conditioned video prediction through extensive experiments on two distinct datasets. Our results show that TextOCVP outperforms several other text-conditioned video prediction methods by effectively leveraging structured object-centric representations, particularly on scenes featuring multiple moving objects. Furthermore, we conduct an in-depth analysis of TextOCVPto verify the effectiveness of object-centric representations for text-guided video prediction. Specifically, we demonstrate that its structured latent space enables accurate and controllable video prediction by aligning language instructions with the corresponding objects, outperforming baselines that rely on holistic scene representations. Beyond generation quality, TextOCVP offers improved interpretability and exhibits strong robustness to novel scene configurations, including varying numbers of objects or previously unseen colors.

In summary, our contributions are as follows:

- We propose TextOCVP, a text-guided video prediction model, featuring a text-conditioned object-centric predictor that integrates textual guidance into the prediction process via a text-to-slot attention mechanism.

- Through extensive evaluations, we show that TextOCVP outperforms other existing text-conditioned video prediction models by leveraging object-centric representations.

- We demonstrate that TextOCVP is controllable, seamlessly adapting to diverse textual instructions, while exhibiting interpretability and robustness to novel setups.

## 2 Related Work

### 2.1 Object-Centric Learning

Representation learning—the ability to extract meaningful features from data—often improves model performance by enhancing its understanding of the input space (Bengio et al., 2013). Object-centric representation

methods aim to parse an image or video into a set of object components in an unsupervised manner. These objects are typically represented as unconstrained embeddings (called slots) (Locatello et al., 2020; Kipf et al., 2022; Singh et al., 2022), patch-based representations (Lin et al., 2020), factored latent vectors (Greff et al., 2019), or object prototypes (Villar-Corrales & Behnke, 2022; Monnier et al., 2020). These methods have demonstrated promise in learning object representations across diverse domains, ranging from synthetic images (Locatello et al., 2020; Lin et al., 2020) to videos (Kipf et al., 2022; Singh et al., 2022), and real-world scenes (Seitzer et al., 2023; Zadaianchuk et al., 2024). The learned object representations benefit downstream tasks, such as reinforcement learning (Mosbach et al., 2025) or visual-question answering (Wu et al., 2023).

## 2.2 Video Prediction

Video prediction (VP) is the task of forecasting the upcoming $T$ video frames conditioned on the preceding $C$ seed frames (Oprea et al., 2020). Several methods have been proposed to address this task, using 2D convolutions (Gao et al., 2022; Chiu et al., 2020), 3D convolutions (Tulyakov et al., 2018), recurrent neural networks (RNNs) (Denton & Fergus, 2018; Villar-Corrales et al., 2022; Wang et al., 2022), transformers (Rakhimov et al., 2021; Ye & Bilodeau, 2022), or diffusion models (Höppe et al., 2022; Ho et al., 2022).

### 2.2.1 Object-Centric Video Prediction

Object-centric VP presents a structured approach that explicitly models the dynamics and interactions of individual objects to forecast future video frames. These methods typically involve three main steps: parsing seed frames into object representations, predicting future object states using a dynamics model, and rendering video frames from the predicted object representations. Various approaches have addressed this task using different architectural priors, such as RNNs (Creswell et al., 2021; Nguyen et al., 2024) or transformers (Wu et al., 2021; 2023; Villar-Corrales et al., 2023; Daniel & Tamar, 2024). Despite promising results, these models are limited to simple deterministic datasets or rely on action-conditioning (Mosbach et al., 2025; Villar-Corrales & Behnke, 2025). In contrast, our model forecasts future frames conditioned on past object slots and text descriptions.

### 2.2.2 Text-Conditioned Video Prediction

Text-conditioned VP models leverage text descriptions to provide appearance, motion and action cues that guide the generation of future frames. This task was first proposed by Hu et al. (2022), who utilized a VQ-VAE to encode images into visual token representations, and modeled the scene dynamics with an axial transformer to jointly process visual tokens with text descriptions. Similarly, approaches like TVP (Song et al., 2024) and MMVG (Fu et al., 2023) address this task using RNNs or masked transformers, respectively. More recently, several methods leverage diffusion models for text-guided VP (Blattmann et al., 2023; Ni et al., 2023; Chen et al., 2024; Xing et al., 2024; Chen et al., 2023; Gu et al., 2024). These approaches encode video frames into discrete token sequences via pretrained quantized autoencoders, and leverage pretrained language models to guide the diffusion-based generation with text features. To further improve temporal coherence and semantic alignment, several diffusion-based methods leverage specialized conditioning strategies (Xing et al., 2024; Chen et al., 2023) or attention mechanisms (Gu et al., 2024), among others. While effective, these models operate on holistic or spatial representations, and require large-scale compute and data. In contrast, TextOCVP adopts a more structured and efficient approach, explicitly modeling object dynamics using slot representations, where each object in the scene is represented by a distinct embedding.

Concurrently with our work, Wang et al. (2025) and Jeong et al. (2025) also combine object-centric learning with autoregressive diffusion models and transformers for text-guided video prediction on simple synthetic datasets. In contrast, we evaluate on more complex robotic simulations, and perform an in-depth analysis of the properties of object-centric representations for text-conditioned video prediction.

## 3 Method

We propose TextOCVP, a novel object-centric model for text-conditioned video prediction. Given an initial reference image $\mathbf{X}_1$ and a text caption $\mathcal{C}$, TextOCVP generates the $T$ subsequent video frames $\hat{\mathbf{X}}_{2:T+1}$, which maintain a similar appearance and structural composition as the reference image, and follow the motion described in the text caption.

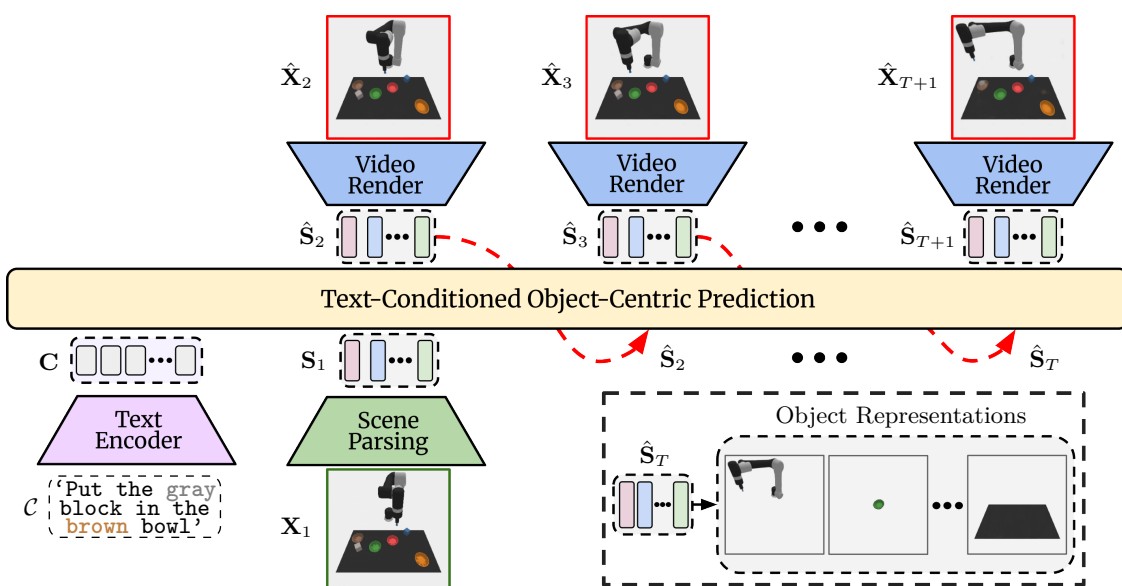

Figure 2: Overview of TextOCVP. Our model parses the reference frame $\mathbf{X}_1$ into object representations $\mathbf{S}_1$. The text-conditioned object-centric predictor models object dynamics, incorporating information from the description $\mathbf{C}$ to predict future object states $\hat{\mathbf{S}}_{2:T+1}$, which can be decoded into video frames $\hat{\mathbf{X}}_{2:T+1}$.

TextOCVP, which is illustrated in Fig. 2, implements an object-centric approach, in which the reference frame $\mathbf{X}_1$ is first decomposed with a scene parsing module (Sec. 3.1) into a set of $N_\mathbf{S}$ $D$-dimensional object representations called slots $\mathbf{S}_1 \in \mathbb{R}^{N_\mathbf{S} \times D}$, where each slot represents a single object in the image. The object slots are fed to a text-conditioned transformer predictor (Sec. 3.2), which jointly models their spatio-temporal relations, and incorporates the textual information from the caption $\mathcal{C}$ as guidance for predicting the future object slots $\hat{\mathbf{S}}_{2:T+1}$. Finally, the predicted slots are decoded to render future video frames (Sec. 3.3).

We propose two different TextOCVP variants, which differ in the underlying object-centric decomposition modules. Specifically, TextOCVP$_{\text{SAVi}}$ leverages SAVi (Kipf et al., 2022), whereas TextOCVP$_{\text{DINO}}$ extends DINOSAUR (Seitzer et al., 2023) for recursive object-centric video decomposition and video rendering.

## 3.1 Scene Parsing

The scene parsing module decomposes a video sequence $\mathbf{X}_{1:\tau}$ into a set of permutation-invariant object representations called slots $\mathbf{S}_{1:\tau} = (\mathbf{S}_1, \ldots, \mathbf{S}_\tau)$, with $\mathbf{S}_t = (\mathbf{s}_t^1, \ldots, \mathbf{s}_t^{N_\mathbf{S}})$, where each slot $\mathbf{s} \in \mathbb{R}^D$ represents a single object. For scene parsing, we adopt the recursive object-centric video decomposition framework from Kipf et al. (2022).

At time step $t$, the corresponding input frame $\mathbf{X}_t$ is encoded with a feature extractor module into a set of $D_h$-dimensional feature maps $\mathbf{h}_t \in \mathbb{R}^{L \times D_h}$ representing $L$ spatial locations. The feature extractor is a convolutional neural network in our TextOCVP$_{\text{SAVi}}$ variant and a DINO-pretrained vision transformer (Caron et al., 2021) in TextOCVP$_{\text{DINO}}$. These feature maps are processed with Slot Attention (Locatello et al., 2020), which updates the previous slots $\mathbf{S}_{t-1}$ based on visual features from the current frame following an iterative attention mechanism. Namely, Slot Attention performs cross-attention, with the attention weights normalized over the slot dimension, thus encouraging the slots to compete to represent parts of the input. It then updates the slots using a Gated Recurrent Unit (Cho et al., 2014) (GRU). Formally, Slot Attention updates the previous slots $\mathbf{S}_{t-1}$ by:

$$\mathbf{A} = \operatorname*{softmax}_{N_\mathbf{S}} \left( \frac{q(\mathbf{S}_{t-1})k(\mathbf{h}_t)^T}{\sqrt{D}} \right) \in \mathbb{R}^{N_\mathbf{S} \times L}, \tag{1}$$

$$\mathbf{S}_t = \text{GRU}(W_t v(\mathbf{h_t}), \mathbf{S}_{t-1}) \quad \text{with} \quad W_{i,j} = \frac{\mathbf{A}_{i,j}}{\sum_{l=1}^{L} \mathbf{A}_{i,l}}, \tag{2}$$

where $k, q$ and $v$ are learned linear projections that map input features and slots into a common dimension. The output of this module is a set of slots $\mathbf{S}_t$ that represents the objects of the input frame.

### 3.2 Text-Conditioned Object-Centric Predictor

Our proposed text-conditioned predictor module, illustrated in Fig. 3, autoregressively forecasts future object states conditioned on the object slots from the reference frame $\mathbf{S}_1$ and a text description $\mathcal{C}$. This design enables TextOCVP to predict temporally consistent object dynamics that are grounded not only in visual context, but also in high-level semantic intent specified via natural language.

To condition the prediction process, the text description $\mathcal{C}$ is encoded into text token embeddings $\mathbf{C}$ using an encoder-only transformer. These tokens provide a global semantic prior guiding predictions at every time step. We experiment with different encoder variants, including vanilla transformer encoder (Vaswani et al., 2017) and pretrained T5 (Raffel et al., 2020). At time step $t$, the predictor receives as input the corresponding text embeddings $\mathbf{C}$, as well as the previous object slots $\mathbf{S}_{1:t}$, which are initially mapped via an MLP to the predictor token dimensionality. Additionally, these tokens are augmented with a temporal positional encoding, which applies the same sinusoidal positional embedding to all tokens from the same time step, thus preserving the inherent permutation-equivariance of the objects.

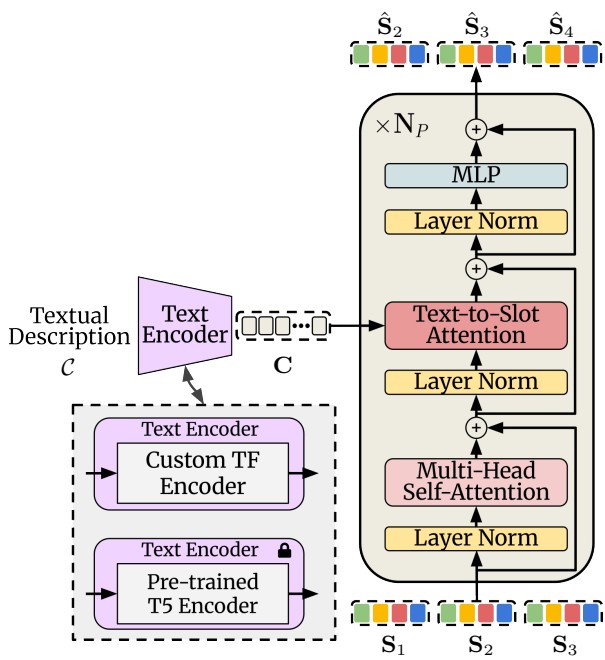

Figure 3: Overview of TextOCVP's text-conditioned object-centric video predictor transformer.

Each layer of our predictor module mirrors the transformer decoder architecture (Vaswani et al., 2017). First, a self-attention layer enables every slot to attend to all other object representations in the sequence, modeling the spatio-temporal relations between objects. Subsequently, a text-to-slot cross-attention layer enhances the slot representations by incorporating relevant features from the text embeddings, guiding the prediction process to align with the motion and dynamics described in the textual caption. Finally, an MLP is applied for each token. This process is repeated in every predictor layer, resulting in the predicted object slots of the subsequent time step $\hat{\mathbf{S}}_{t+1}$. Furthermore, we apply a residual connection from $\mathbf{S}_t$ to $\hat{\mathbf{S}}_{t+1}$, which improves the prediction temporal consistency. This process is repeated autoregressively to obtain slot predictions for $T$ subsequent time steps.

### 3.3 Video Rendering

The video rendering module decodes the predicted slots $\hat{\mathbf{S}}_t$ to render the corresponding video frame $\hat{\mathbf{X}}_t$. We leverage two variants of the video rendering module, for our TextOCVP$_{\text{SAVi}}$ and TextOCVP$_{\text{DINO}}$ variants.

**TextOCVP$_{\text{SAVi}}$ Decoder**     This variant independently decodes each slot in $\hat{\mathbf{S}}_t$ with a CNN-based Spatial Broadcast Decoder (Watters et al., 2019), rendering an object image $\mathbf{o}_t^n$ and mask $\mathbf{m}_t^n$ for each slot $\mathbf{s}_t^n$. The object masks are normalized across the slot dimension, and the representations are combined via a weighted sum to render video frames:

$$\hat{\mathbf{X}}_t = \sum_{n=1}^{N_{\mathbf{S}}} \mathbf{o}_t^n \cdot \tilde{\mathbf{m}}_t^n \quad \text{with} \quad \tilde{\mathbf{m}}_t^n = \underset{N_{\mathbf{S}}}{\text{softmax}}(\mathbf{m}_t^n). \tag{3}$$

**TextOCVP$_{\textbf{DINO}}$ Decoder** This decoder variant decodes the object slots in two distinct stages. First, following DINOSAUR (Seitzer et al., 2023), an MLP-based Spatial Broadcast Decoder (Watters et al., 2019) is used to generate object features along with their corresponding masks. Similar to the TextOCVP$_{\text{SAVi}}$ decoder, the object masks are normalized and combined with the object features in order to reconstruct the encoded features $\hat{\mathbf{h}}_t \in \mathbb{R}^{L \times D_h}$. In the second stage, the reconstructed features $\hat{\mathbf{h}}_t$ are arranged into a grid format and processed with a CNN decoder to generate the corresponding video frame $\hat{\mathbf{X}}_t$.

### 3.4 Training and Inference

Our proposed TextOCVP is trained in two different stages.

**Object-Centric Learning** We first train the scene parsing and video rendering modules for parsing video frames into object-centric representations by minimizing a reconstruction loss. In the TextOCVP$_{\text{SAVi}}$ variant ($\mathcal{L}_{\text{SAVi}}$), these modules are trained by reconstructing the input images, whereas in TextOCVP$_{\text{DINO}}$ ($\mathcal{L}_{\text{DINO}}$) they are trained by jointly minimizing an image and a feature reconstruction loss:

$$\mathcal{L}_{\text{SAVi}} = \frac{1}{T}\sum_{t=1}^{T}||\hat{\mathbf{X}}_t - \mathbf{X}_t||_2^2, \qquad \mathcal{L}_{\text{DINO}} = \frac{1}{T}\sum_{t=1}^{T}\left(||\hat{\mathbf{X}}_t - \mathbf{X}_t||_2^2 + ||\hat{\mathbf{h}}_t - \mathbf{h}_t||_2^2\right). \tag{4}$$

**Predictor Training** Given the pretrained scene parsing and rendering modules, we train our TextOCVP predictor for text-conditioned video prediction using a dataset containing paired videos and text descriptions. Namely, given the object representations from a reference frame $\mathbf{S}_1$ and the textual embeddings $\mathbf{C}$, the predictor forecasts subsequent object slots $\hat{\mathbf{S}}_2$, which are decoded into a predicted video frame $\hat{\mathbf{X}}_2$. This process is repeated autoregressively, i.e. the predicted slots are appended to the input in the next time step, in order to generate the set of slots for the subsequent $T$ time steps. This autoregressive training, in contrast to teacher forcing, enforces our predictor to operate with imperfect inputs, leading to better modeling of long-term dynamics at inference time. TextOCVP predictor is trained by minimizing the following loss:

$$\mathcal{L}_{\text{TextOCVP}} = \frac{1}{T}\sum_{t=2}^{T+1}\left(\lambda_{\text{Img}}\mathcal{L}_{\text{Img}} + \lambda_{\text{Slot}}\mathcal{L}_{\text{Slot}}\right), \tag{5}$$

$$\text{with } \mathcal{L}_{\text{Img}} = ||\hat{\mathbf{X}}_t - \mathbf{X}_t||_2^2 \text{ and } \mathcal{L}_{\text{Slot}} = ||\hat{\mathbf{S}}_t - \mathbf{S}_t||_2^2, \tag{6}$$

where $\mathcal{L}_{\text{Img}}$ measures the future frame prediction error, and $\mathcal{L}_{\text{Slot}}$ enforces the alignment of the predicted object slots with the actual inferred object-centric representations.

**Inference** At inference time, TextOCVP receives as input a single reference frame and a language instruction. Our model parses the seed frame into object slots and autoregressively predicts future object states and video frames conditioned on the given textual description. By modifying the language instruction, TextOCVP can generate a new sequence continuation that performs the specified task while preserving a consistent scene composition.

## 4 Experiments

### 4.1 Experimental Setup

#### 4.1.1 Datasets

We evaluate TextOCVP for text-conditioned video prediction on the CATER and CLIPort datasets:

*CATER* (Girdhar & Ramanan, 2020) is a dataset of long video sequences showing 3D objects in motion, each paired with a descriptive caption. We used the CATER-hard variant (Hu et al., 2022), containing 30,000 sequences with $64 \times 64$ frames featuring two to eight objects, two of which follow the scripted motion.

*CLIPort* (Shridhar et al., 2022) is a robot manipulation dataset featuring video-caption pairs. We employ 21,000 $336 \times 336$ sequences of the Put-Block-In-Bowl task, where each scene contains six objects, either a block or a bowl, on a 2D table plane, and a robot arm. The caption describes placing a block into a bowl.

Table 1: Quantitative evaluation on CATER and CLIPort datasets for prediction horizons of $T = 9$ and $T = 19$. TextOCVP outperforms the baselines. Best two results are shown in bold and underlined, respectively.

(a) Quantitative evaluation on CATER.

| Method | $\mathbf{CATER}_{1 \to 9}$ | | | | $\mathbf{CATER}_{1 \to 19}$ | | | |
|---|---|---|---|---|---|---|---|---|
| | PSNR↑ | SSIM↑ | LPIPS↓ | JEDi↓ | PSNR↑ | SSIM↑ | LPIPS↓ | JEDi↓ |
| OCVP (Villar-Corrales et al., 2023) | 29.08 | 0.874 | 0.078 | 4.16 | 28.11 | 0.854 | 0.101 | 8.08 |
| Non-OC | 29.68 | 0.874 | 0.092 | 3.04 | 28.39 | 0.849 | 0.112 | 8.62 |
| SEER (Gu et al., 2024) | 22.05 | 0.723 | 0.245 | 11.23 | 16.05 | 0.535 | 0.299 | 17.29 |
| MAGE (Hu et al., 2022) | **34.91** | 0.877 | 0.108 | 3.46 | **34.76** | 0.871 | 0.111 | 5.88 |
| TextOCVP (Ours) | 32.98 | **0.922** | **0.036** | **2.16** | 31.29 | **0.902** | **0.044** | **5.09** |

(b) Quantitative evaluation on CLIPort.

| Method | $\mathbf{CLIPort}_{1 \to 9}$ | | | | $\mathbf{CLIPort}_{1 \to 19}$ | | | |
|---|---|---|---|---|---|---|---|---|
| | PSNR↑ | SSIM↑ | LPIPS↓ | JEDi↓ | PSNR↑ | SSIM↑ | LPIPS↓ | JEDi↓ |
| Non-OC | 23.44 | 0.901 | 0.184 | 8.13 | 20.14 | 0.872 | 0.210 | 13.23 |
| SEER (Gu et al., 2024) | 21.01 | 0.887 | 0.141 | 6.80 | 11.30 | 0.622 | 0.331 | 8.29 |
| MAGE$_{\text{DINO}}$ (Hu et al., 2022) | 23.72 | 0.940 | 0.064 | 2.11 | 22.27 | **0.931** | **0.075** | 2.59 |
| TextOCVP (Ours) | **26.99** | **0.950** | **0.062** | **1.36** | **23.88** | **0.931** | 0.078 | **2.23** |

### 4.1.2 Baselines

We benchmark TextOCVP against established text-guided video prediction baselines and analyse key architectural design choices. To assess the impact of text conditioning, we compare TextOCVP with OCVP-Seq (Villar-Corrales et al., 2023), an unconditional object-centric prediction model. To evaluate the role of object-centric representations, we introduce a TextOCVP variant (*Non-OC*) replacing slot representations with a single holistic embedding. We further compare TextOCVP with three transformer- or diffusion-based text-conditioned prediction models: MAGE (Hu et al., 2022), MAGE$_{\text{DINO}}$, and SEER (Gu et al., 2024). Further details on baseline models are provided in Appendix C.

### 4.1.3 Implementation Details

All models are implemented in PyTorch and trained on a single NVIDIA A6000 (48Gb) GPU. TextOCVP$_{\text{SAVi}}$ closely follows Kipf et al. (2022) for scene parsing and video rendering. TextOCVP$_{\text{DINO}}$ uses DINOv2 (Oquab et al., 2024) as image encoder, a four-layer MLP Spatial-Broadcast Decoder (Watters et al., 2019) to decode slots into object features and masks, and a CNN decoder to map the reconstructed scene features back to images. On CATER, we use the TextOCVP$_{\text{SAVi}}$ variant with eight 128-dimensional object slots, whereas on CLIPort we employ the TextOCVP$_{\text{DINO}}$ variant with ten 128-dimensional slots. Our predictor module is an eight-layer transformer with 512-dimensional tokens, eight attention heads, and a hidden dimension of 1024. Further experimental details—including datasets, baselines, evaluation metrics, and implementation details—are provided in Appendices B-E.

### 4.2 Results

### 4.2.1 CATER Results

On CATER, we train the models to predict nine future frames given a single reference frame and a text caption. In Table 1a we report quantitative evaluations on CATER using the same setting as in training, i.e. predicting $T = 9$ frames, as well as when predicting $T = 19$ future frames. In both settings, TextOCVP outperforms all other models, demonstrating superior perceptual quality.

Fig. 4a qualitatively compares TextOCVP and MAGE on CATER. TextOCVP generates a sequence that exhibits coherent object trajectories and accurate motion grounded in the input textual instruction. This alignment is made possible by the explicit object-centric design, which disentangles the scene into structured object representations and allows the model to reason about each object's dynamics independently. In

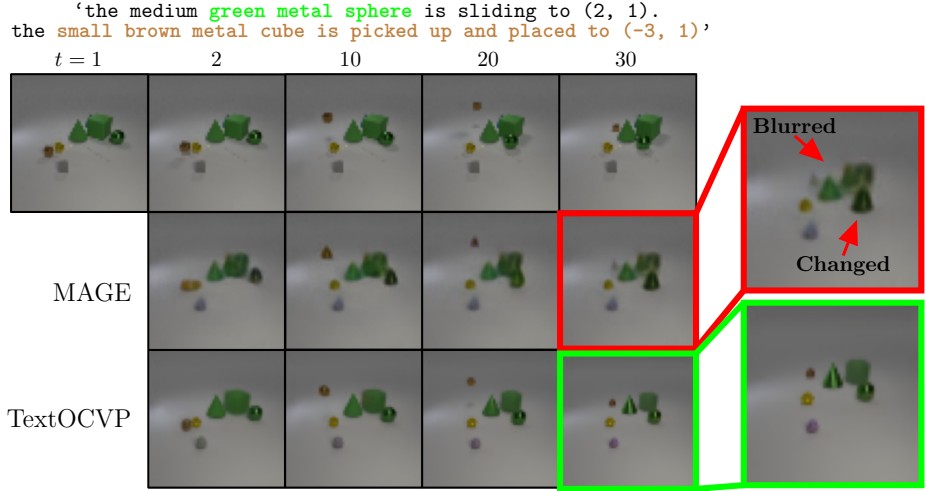

(a) Text-guided video prediction result on CATER. Top row depicts the ground truth frames. TextOCVP predicts sharp and accurate frames, whereas MAGE blurs and misses objects.

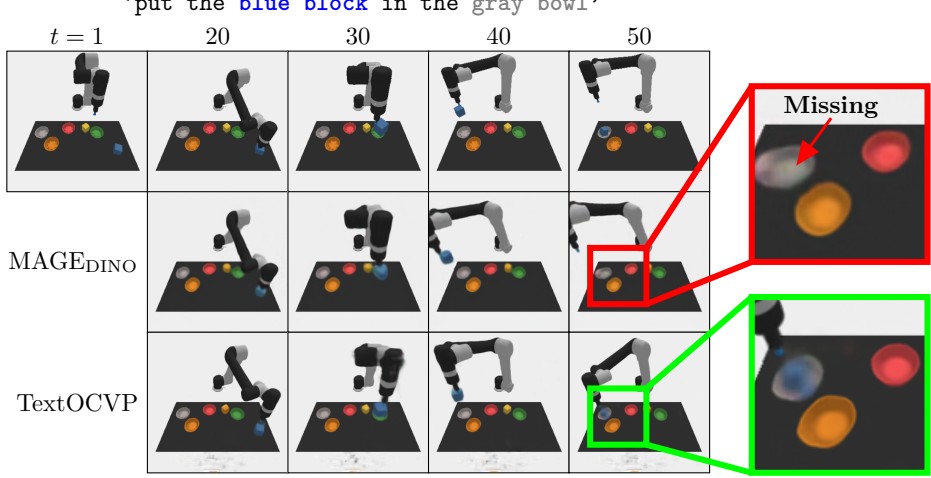

(b) Qualitative evaluation on CLIPort for text-driven video prediction. Top row depicts the ground truth frames. TextOCVP successfully completes the pick-and-place task, whereas in MAGE$_{\text{DINO}}$ the moved block disappears.

Figure 4: Qualitative comparison of TextOCVP and baseline methods on text-guided video prediction.

contrast, MAGE predictions feature multiple errors and artifacts, including missing objects, blurry contours, and significant changes on object shapes. These issues arise from the difficulty of modeling object-specific dynamics in pixel or token space without explicit object-level abstraction.

These results demonstrate the importance of structured, object-centric representations in capturing the compositional and controllable nature of physical scenes, which is especially critical in tasks involving fine-grained object manipulations.

### 4.2.2 CLIPort Results

Table 1b reports quantitative results for text-guided video prediction on CLIPort. TextOCVP outperforms all baselines when evaluated on the same setting as in training (i.e. $1 \rightarrow 9$). For longer prediction horizons, both MAGE$_{\text{DINO}}$ and TextOCVP perform well, with TextOCVP achieving the best scores on JEDi (Luo et al., 2025)—a perceptual video metric evaluating motion quality and realism.

Table 2: Ablation studies on key design choices in TextOCVP, including predictor depth (2a), text encoder choice (2b), residual connections (2c), and slot count (2d).

(a) Effect of number of layers ($N_P$) in predictor module.

| $N_P$ | CATER$_{1\to9}$ | | |
| --- | --- | --- | --- |
| | SSIM↑ | LPIPS↓ | JEDi↓ |
| 2 | 0.908 | 0.045 | 2.26 |
| 4 | 0.911 | 0.043 | 2.20 |
| 8 | **0.922** | **0.036** | **2.16** |

(b) Impact of different text encoders, including a transformer encoder (TF) and two T5 variants, on CATER and CLIPort.

| Text Enc. | CATER$_{1\to19}$ | | | CLIPort$_{1\to19}$ | | |
| --- | --- | --- | --- | --- | --- | --- |
| | SSIM↑ | LPIPS↓ | JEDi↓ | SSIM↑ | LPIPS↓ | JEDi↓ |
| TF | **0.903** | 0.045 | 5.39 | - | - | - |
| Frozen T5 | 0.902 | 0.044 | **5.09** | **0.931** | **0.078** | **2.23** |
| FT T5 | 0.901 | **0.043** | 5.14 | 0.928 | 0.082 | 2.67 |

(c) Effect of residual connection in predictor.

| Residual | CLIPort$_{1\to9}$ | | |
| --- | --- | --- | --- |
| | SSIM↑ | LPIPS↓ | JEDi↓ |
| ✗ | 0.946 | 0.066 | 1.58 |
| ✓ | **0.950** | **0.062** | **1.36** |

(d) Effect of the number of object slots ($N_S$).

| # Slots | CLIPort$_{1\to9}$ | | |
| --- | --- | --- | --- |
| | SSIM↑ | LPIPS↓ | JEDi↓ |
| 8 | 0.946 | 0.079 | 5.62 |
| 10 | **0.950** | **0.062** | **1.36** |

In our qualitative evaluations, we observe that TextOCVP generates the most accurate sequence predictions given the reference frame and language instruction. Fig. 4b compares TextOCVP with MAGE$_{DINO}$ over a long prediction horizon of 50 frames, corresponding to the full completion of the given task. MAGE$_{DINO}$ fails to complete the task outlined in the textual description, as it misses the target block after several prediction time steps. In contrast, TextOCVP successfully completes the instructed task, maintaining coherent object trajectories and consistent scene dynamics throughout the sequence.

Nevertheless, we observe that TextOCVP often generates visual artifacts and lacks fine-grained textures. While these imperfections can affect its quantitative scores, TextOCVP's object-centric structure enables robust long-term prediction and precise instruction-following behavior.

## 4.3 Model Analysis

### 4.3.1 Ablation Studies

We perform several ablation studies to support and validate the architectural choices of our model components and their impact on TextOCVP's video prediction performance. The results are presented in Table 2, and analysed below across four main design axes. Further results and analysis are provided in Appendix F.

**Number of Layers (Table 2a)** We study the effect of increasing the number of transformer layers ($N_P$) in the predictor module. We observe that prediction quality improves with model depth, with $N_P = 8$ yielding the best performance across all metrics on the CATER dataset. This suggests that deeper predictor models enable more accurate modeling of object dynamics. However, we did not explore beyond eight layers due to the substantial increase in computational cost, training time and parameter count.

**Residual Connection (Table 2c)** We evaluate the effect of introducing a residual connection in the predictor module, where the updated slot representation is defined as $\hat{\mathbf{S}}_{t+1} = \mathbf{S}_t + f_{\text{pred}}(\mathbf{S}_t)$. Adding this residual path improves performance across all metrics on the CLIPort dataset, particularly in improving the temporal consistency of predicted slots. This result is consistent with prior findings (Villar-Corrales et al., 2023), which show that residual updates are beneficial for iterative refinement in structured latent spaces.

**Text Encoder (Table 2b)** We evaluate the performance of three different text encoders, including a lightweight transformer trained from scratch (TF), a frozen T5 encoder, and a fine-tuned (FT) T5. While all three perform similarly on CATER, the frozen T5 text encoder achieves the best results on CLIPort, especially on perceptual metrics such as JEDi and LPIPS. Interestingly, fine-tuning the T5 encoder led to

Table 3: Quantitative evaluation on two CLIPort test sets with unseen colors and larger number of objects. We report the absolute video prediction performance and the relative drop (in parentheses) compared to the original evaluation setting. Best result is marked in bold.

(a) Evaluation on CLIPort test set with objects of colors unseen during training.

| Method | $\textbf{CLIPort}_{1\to 9}$ | | $\textbf{CLIPort}_{1\to 19}$ | |
|---|---|---|---|---|
| | SSIM↑ | LPIPS↓ | SSIM↑ | LPIPS↓ |
| MAGE$_{\text{DINO}}$ (Hu et al., 2022) | 0.935 (-0.5%) | 0.076 (-19%) | 0.924 (-0.7%) | 0.087 (-16%) |
| TextOCVP (Ours) | **0.946 (-0.4%)** | **0.066 (-6.4%)** | **0.927 (-0.4%)** | **0.083 (-6.4%)** |

(b) Evaluation on CLIPort test set with a larger object count than training scenes.

| Method | $\textbf{CLIPort}_{1\to 9}$ | | $\textbf{CLIPort}_{1\to 19}$ | |
|---|---|---|---|---|
| | SSIM↑ | LPIPS↓ | SSIM↑ | LPIPS↓ |
| MAGE$_{\text{DINO}}$ (Hu et al., 2022) | 0.929 **(-1.2%)** | 0.088 (-37.5%) | 0.920 (-1.2%) | 0.094 (-25.3%) |
| TextOCVP (Ours) | **0.936** (-1.5%) | **0.076 (-22.6%)** | **0.921 (-1.1%)** | **0.090 (-15.4%)** |

slightly degraded performance, likely due to overfitting or interference with the pretrained representations. We therefore use the frozen T5 encoder in all main experiments to balance performance and model efficiency.

**Number of Slots (Table 2d)**     Finally, we explore the impact of varying the number of slots ($N_{\textbf{S}}$) used to represent the scene. Although each CLIPort scene can be described with eight slots—representing six objects, the robot arm, and the background—we find that using ten slots results in notably better performance. This observation suggests that the additional slots can function as internal registers, supporting attention routing or acting as a form of cache that aids with internal model computations (Darcet et al., 2024).

### 4.3.2  Model Robustness and Generalization

We evaluate the robustness and generalization of TextOCVP and MAGE$_{\text{DINO}}$ on two CLIPort evaluation sets: one involving color variations in the text instructions that were not encountered during training (unseen-color), and another featuring scenes with more objects than observed during training, eight instead of six (more-objects). These experiments assess model robustness under distribution shifts and test the ability to generalize to unseen visual-linguistic combinations and novel scene configurations. Results are summarized in Table 3, reporting both the absolute performance and the relative drop compared to the standard (seen-color or six-object) settings.

On the unseen-color benchmark (Table 3a), TextOCVP consistently outperforms MAGE$_{\text{DINO}}$ across all metrics and prediction horizons, showing significantly smaller degradation in perceptual similarity (LPIPS). This indicates stronger robustness to novel color-object combinations. In the more-objects setting (Table 3b), TextOCVP again maintains superior performance with a notably smaller decline in LPIPS, demonstrating resilience to higher scene complexity and unseen object counts. Qualitative examples in Appendix F.6 further support these findings.

Overall, these results highlight the advantages of structured object-centric representations, which provide greater generalization and robustness in video prediction tasks, particularly in contrast to holistic scene representations that struggle with novel scene compositions.

### 4.3.3  Interpretability

Fig. 5a visualizes the text-to-slot attention weights—averaged across heads—for different slots in a CATER sequence, highlighting how textual instructions guide the model's predictions. Slots corresponding to described objects attend strongly to the relevant tokens in the language, such as actions or target coordinates. To further dissect this behavior, Fig. 5b shows the attention distribution of individual text-to-slot attention

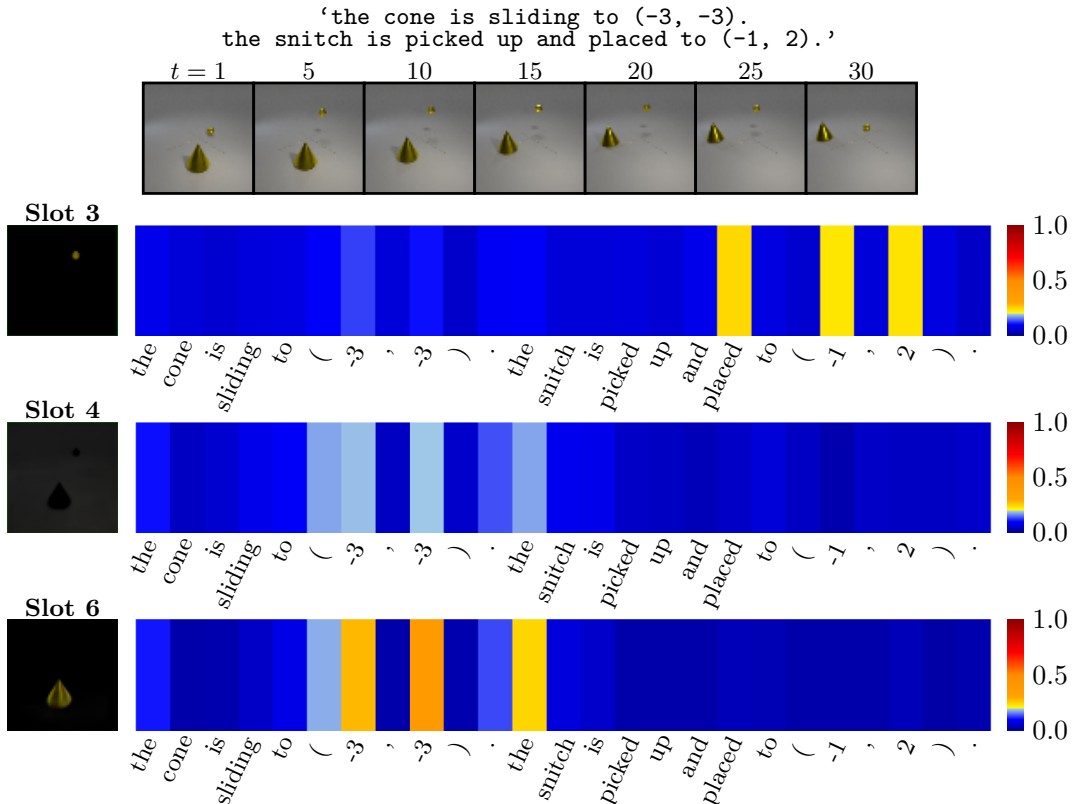

(a) Text-to-slot attention weights, averaged across heads, for different objects in a CATER sequence.

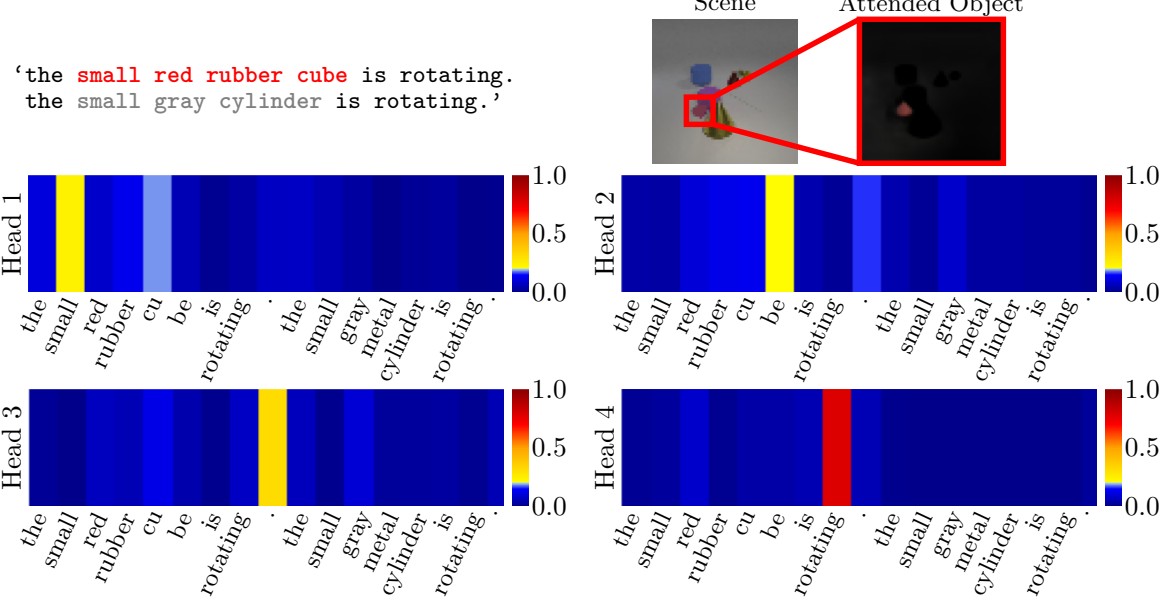

(b) Text-to-slot attention weights between an object slot and the textual tokens for four different heads.

Figure 5: Visualization of text-to-slot attention in TextOCVP. (a) Slots attend to relevant text tokens, grounding objects to their described motions. (b) Distinct attention heads focus on complementary textual cues, such as object attributes and actions.

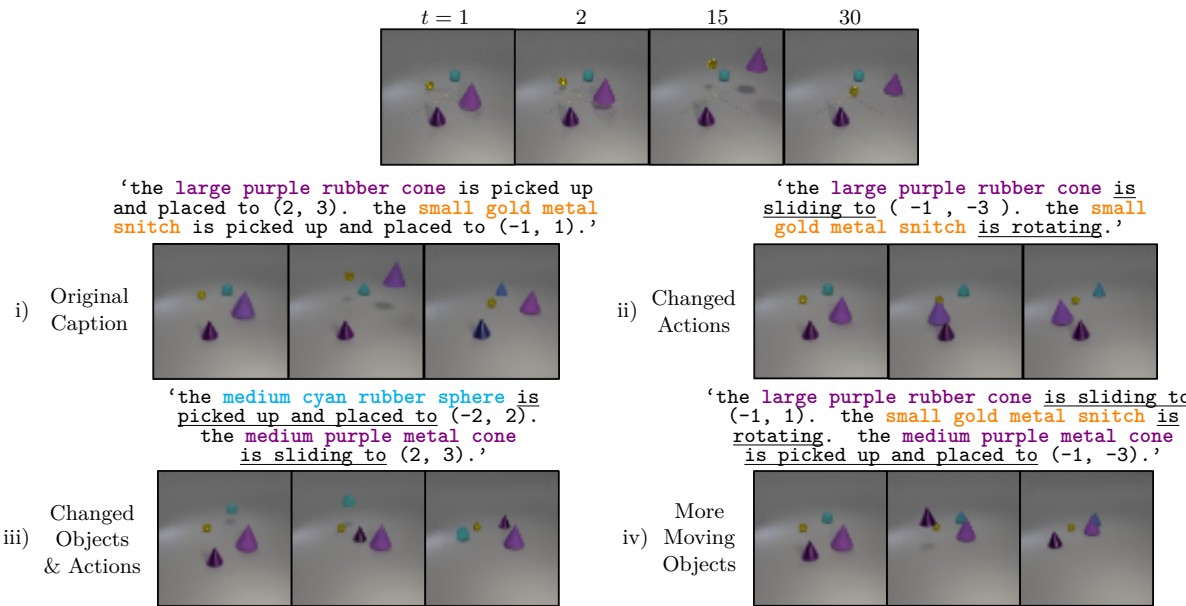

(a) Qualitative evaluation of TextOCVP's controllability on CATER. Top row shows the ground truth sequence. We underline the changed actions with respect to the original caption. TextOCVP demonstrates fine-grained control by predicting different sequence continuations from the same reference frame, each conditioned on a different instruction.

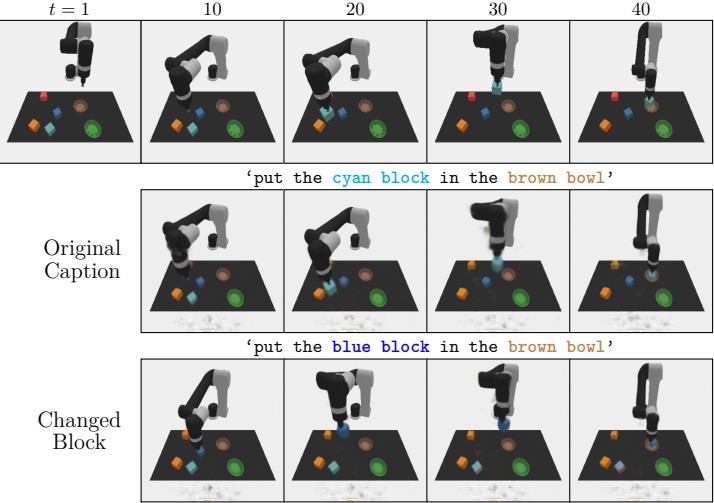

(b) Qualitative evaluation of TextOCVP's controllability on CLIPort. Top row shows ground truth frames. TextOCVP correctly generates sequences where the robot picks up the correct block and places it into the specified bowl.

Figure 6: Qualitative evaluation of TextOCVP's controllability on CATER and CLIPort datasets.

heads for a slot representing a rotating red cube. Different heads specialize in distinct attributes of the text, including the object's shape (Head 2), size (Head 1), and the described motion (Head 4).

Together, these results show that the text-to-slot attention mechanism effectively aligns textual information with object-centric representations. This enables accurate video prediction conditioned on natural language, and provides a degree of interpretability by revealing which parts of text influence each object representation.

### 4.3.4 Controllability

A key objective of text-guided video prediction is to provide fine-grained control over the prediction process via language instructions that specify the relevant objects and their actions.

Fig. 6a illustrates TextOCVP's controllability on CATER. Starting from the same reference frame, we generate multiple sequence continuations by varying the natural language instruction. These variations include changing the target objects and their actions, as well as instructions that specify a greater number of actions than seen during training. As shown in Fig. 6a, TextOCVP successfully identifies the relevant objects and executes the described actions. Notably, it distinguishes between two nearly identical purple cones, despite their identical shapes and color, and generates sequences consistent with the specified motions.

A similar experiment on CLIPort is shown in Fig. 6b. Given a single frame with multiple colored blocks and bowls, modifying the instruction determines which block the robot arm picks and the destination bowl. In both cases, TextOCVP correctly selects the specified block, places it in the instructed bowl, and adapts the arm trajectory to the described action.

These results show that key object attributes, such as size or color, are effectively captured through the text-to-slot attention mechanism. This enables accurate, per-object motion forecasting and highlights the benefit of combining object-centric representations with language guidance for controllable video prediction. Further quantitative and qualitative evaluations of TextOCVP's controllability are provided in Appendices F.5 and F.10, respectively.

## 5 Conclusion

In this work, we presented TextOCVP, an object-centric model for text-conditioned video prediction. Given a single input image and a natural language description, TextOCVP generates future frames by parsing the scene into slot-based object representations and modeling their dynamics conditioned on the text instruction. This is accomplished through a text-conditioned object-centric transformer that predicts future object states by modeling spatio-temporal relationships between objects while incorporating textual guidance. Through extensive evaluations, we demonstrated that TextOCVP outperforms other existing approaches for text-driven video prediction from a single frame, highlighting our model's ability to predict over long prediction horizons and adapt its predictions based on the provided description. Moreover, we validated our architectural choices through ablation studies, highlighting the importance of combining textual and object-centric information, and demonstrating strong robustness and interpretability. With its structured latent space and superior controllability, TextOCVP offers a promising step toward controllable object-centric manipulation in simulated robotic environments, supporting more efficient planning, reasoning and decision-making.

## Acknowledgment

This work was funded by grant BE 2556/16-2 (Research Unit FOR 2535 Anticipating Human Behavior) of the German Research Foundation (DFG). Computational resources were provided by the German AI Service Center WestAI.

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

# A    Limitations and Future Work

## A.1    Limitations

While TextOCVP demonstrates promising results for text-guided object-centric video prediction, it presents some limitations, which we plan to address in future work:

**Prediction Artifacts**    TextOCVP occasionally generates artifacts in the predicted frames, such as blurriness, inconsistent object appearances, lack of textured details, or visual artifacts in the background. We believe that these limitations stem from the video rendering module, which might lack the representational power to reconstruct precise image details from the object-centric latent space representation.

**Limited Temporal Consistency**    While TextOCVP produces plausible predictions, we observe that its temporal consistency can degrade when forecasting for long prediction horizons ($T > 30$), occasionally resulting in object jittering or instability. We attribute this limitation to the fact that TextOCVP is trained to predict only up to nine future frames ($T = 9$) and only optimizing reconstruction losses, which do not penalize such temporal inconsistencies.

**Inherent Limitations of Slot-based Models**    Slot-based object-centric models, while effective for disentangling scene structure and producing interpretable object-level representations, come with several inherent limitations. They often struggle to represent fine textures, objects with intricate geometry, small visual details, or tightly interlocking and highly deformable objects, as these phenomena exceed the granularity and capacity of a single slot embedding. In such cases, relevant information is often spread across multiple slots, leading to oversegmentation issues, or compressed into an overly coarse latent, leading to imperfect and blurred reconstructions, lack of representation detail, or unstable object binding. More fundamentally, current slot-based architectures typically operate at a single spatial scale, which prevents them from naturally encoding part-whole hierarchies, capturing object structure at different levels of detail, or representing complex compositional structures. Addressing these challenges through hierarchical object-centric parsing, adaptive slot resolution, or richer generative decoders remains an important future direction for object-centric learning and video prediction.

## A.2    Future Work

In future work we aim to address these limitations. Our TextOCVP model is designed with a modular architecture, enabling for both improvements and flexible swapping of components—such as the parsing, predictor, or video rendering modules—to seamlessly improve the entire pipeline.

To address the prediction artifacts, we plan to extend our TextOCVP framework with more powerful decoder modules, such as autoregressive transformers (Singh et al., 2022) or diffusion models (Jiang et al., 2023), as well as scale our predictor module. Furthermore, we plan to incorporate temporal discriminators (Clark et al., 2020) to improve the temporal consistency of the predicted video frames.

We believe that exploring these architectural modifications can overcome the aforementioned limitations and will enable the scaling of TextOCVP to real-world robotic environments.

# B    Implementation Details

We employ TextOCVP$_{\text{SAVi}}$ for the experiments on CATER and TextOCVP$_{\text{DINO}}$ for experiments on CLIPort. Below we discuss the implementation details for each of these variants.

## B.1    TextOCVP$_{\text{DINO}}$

The TextOCVP$_{\text{DINO}}$ variant consists of our proposed text-conditioned predictor module and an object-centric decomposition module that extends the DINOSAUR (Seitzer et al., 2023) framework (Extended DINOSAUR) for recursive object-centric video decomposition and video rendering.

**Text-Conditioned Predictor** The predictor is composed of $N_{\mathrm{P}} = 8$ identical layers, each containing 8-head attention mechanisms and an MLP with a single hidden layer of dimension 1024 and a ReLU activation function. Furthermore, the predictor uses an embedding dimensionality of 512, context window size of ten frames, and applies a residual connection from the predictor input to its output.

**Text Encoder** TextOCVP$_{\mathrm{DINO}}$ leverages a pretrained and frozen small version of T5 encoder (Raffel et al., 2020), which consists of six T5 blocks. This text encoder uses a vocabulary with size 32,128.

**Scene Parsing** The scene parsing module generates $N_{\mathbf{S}} = 10$ slots of dimension 128. As feature extractor, we use DINOv2 ViT-Base (Oquab et al., 2024), featuring 12 layers, using a patch size of 14, and producing patch features with dimension $D_h = 768$. The Slot Attention corrector module processes the first video frame with three iterations in order to obtain a good initial object-centric decomposition, and a single iteration for subsequent frames, which suffices to recursively update the slot representation. The initial object slots $\mathbf{S}_0$ are randomly sampled from a Gaussian distribution with learned mean and covariance. We use a single Transformer encoder block as the transition function, which consists of four attention heads and an MLP with a hidden dimension of 512.

**Video Rendering** The video rendering module consists of two distinct decoders. First, a four-layer MLP-based Spatial Broadcast Decoder (Watters et al., 2019) with hidden dimension 1024 reconstructs the patch features from the slots. Then, a CNN-based decoder reconstructs full-resolution images from these features. It consists of four convolutional layers, each using $3 \times 3$ kernels, a ReLU activation function and bilinear upsampling. A final convolutional layer and bilinear interpolation are applied to map the outputs to the RGB channels and spatial dimensions of the image.

**Training** We train our model for object-centric decomposition using video sequences of length five frames for 1000 epochs. We use batch size of 16, the Adam optimizer (Kingma & Ba, 2015), and a base learning rate of $4 \times 10^{-4}$, which is linearly warmed-up for the first 10000 steps, followed by cosine annealing for the remaining of the training process. Moreover, we clip the gradients to a maximum norm of 0.05. The predictor module is trained given the frozen and pretrained object-centric decomposition model for 700 epochs to predict the subsequent nine frames using a single seed frame. The predictor is trained using the same hyper-parameters as for object-centric decomposition. In the predictor loss function $\mathcal{L}_{\mathrm{TextOCVP}}$, we set $\lambda_{\mathrm{Img}} = 1$ and $\lambda_{\mathrm{Slot}} = 1$.

### B.2 TextOCVP$_{\mathbf{SAVi}}$

TextOCVP$_{\mathrm{SAVi}}$ uses the same text-conditioned predictor and text-encoder architectures as TextOCVP$_{\mathrm{DINO}}$, but employs SAVi (Kipf et al., 2022) as the object-centric decomposition module.

**Scene Parsing** The scene parsing module generates $N_{\mathbf{S}} = 8$ slots of dimension 128. Following Kipf et al. (2022), we use as feature extractor a four-layer CNN with ReLU activation function, where each convolutional layer features 32 $5 \times 5$ kernels, stride $= 1$, and padding $= 2$. The Slot Attention corrector follows the same structure as in TextOCVP$_{\mathrm{DINO}}$.

**Video Rendering** Following Kipf et al. (2022), we utilize a CNN-based Spatial Broadcast Decoder (Watters et al., 2019) with four convolutional layers with 32 kernels of size $5 \times 5$ , stride $= 1$, and padding $= 2$. A final convolutional layer maps from the hidden 32-channel representation to four output channels (RGB + alpha mask).

**Training** We train our model for object-centric decomposition using video sequences of length ten frames for 1000 epochs, using batch size of 64, and an initial learning rate of $10^{-4}$, which is warmed up for 2500 steps, followed by cosine annealing for the remaining of the training process. Moreover, we clip the gradients to a maximum norm of 0.05. The predictor module is trained given the frozen and pretrained object-centric decomposition model for 1400 epochs to predict the subsequent nine frames using a single seed frame. The

Table 4: Number of learnable parameters in TextOCVP and baselines for experiments on CLIPort.

| Model | # Parameters |
|---|---|
| TextOCVP | 33.76M |
| Non-OC | 34.16M |
| $\text{MAGE}_{\text{DINO}}$ | 32.11M |
| SEER | 405.89M |

predictor is trained using the same hyper-parameters as for object-centric decomposition. In the predictor loss function $\mathcal{L}_{\text{TextOCVP}}$, we set $\lambda_{\text{Img}} = 1$ and $\lambda_{\text{Slot}} = 1$.

## C  Baselines

We employ five different baselines to compare against our TextOCVP model for the task of text-conditioned video prediction on the CATER and CLIPort datasets. To emphasize the importance of incorporating textual information, we include a comparison with OCVP-Seq (Villar-Corrales et al., 2023), a recent object-centric video prediction model that does not utilize text conditioning. Additionally, we evaluate a non-object-centric TextOCVP variant (*Non-OC*) that processes the input image into a single high-dimensional slot representation, instead of multiple object-centric slots, thus allowing us to evaluate the effect of object-centric representations. Moreover, we compare TextOCVP with three popular text-conditioned video prediction baselines that do not incorporate object-centricity: the transformer-based MAGE (Hu et al., 2022) model and its $\text{MAGE}_{\text{DINO}}$ variant, and the diffusion-based SEER (Gu et al., 2024) model. We train these baselines on CATER and CLIPort closely following the original implementation details[1][2].

Table 4 lists the number of learnable parameters in our proposed TextOCVP as well as for the baseline models on CLIPort. TextOCVP, Non-OC and $\text{MAGE}_{\text{DINO}}$ employ a comparable number of parameters, thus ensuring a fair comparison. SEER employs a pretrained latent diffusion model, which already requires a significantly larger number of parameters, and adapts it for the task of text-guided image-to-video generation.

### C.1  MAGE and MAGE$_{\text{DINO}}$

MAGE is an autoregressive text-guided video prediction framework that utilizes a VQ-VAE (Van Den Oord et al., 2017) encoder-decoder architecture to learn efficient visual token representations. A cross-attention module aligns textual and visual embeddings to produce a spatially-aligned motion representation termed Motion Anchor (MA), which is fused with visual tokens via an axial transformer for video generation. For experiments on CATER, we use a codebook size of $512 \times 256$ with a downsampling ratio of four, whereas on CLIPort we use a codebook size of $512 \times 1024$.

To ensure a fair comparison with TextOCVP on CLIPort, we replace MAGE's standard CNN encoder and decoder with the DINOv2 ViT encoder and CNN decoder used in our TextOCVP model. We refer to this modified version as $\text{MAGE}_{\text{DINO}}$. Table 5 presents a comparison between the original MAGE model and $\text{MAGE}_{\text{DINO}}$ on CLIPort. The results demonstrate that $\text{MAGE}_{\text{DINO}}$ significantly outperforms the original variant, enabling a fair comparison with TextOCVP and other baselines on this benchmark.

MAGE and $\text{MAGE}_{\text{DINO}}$ share several architectural similarities with our proposed approach—using similar encoder and decoder modules, text-conditioning, and an autoregressive transformer for prediction. However, these models differ from TextOCVP in two fundamental ways:

**Scene representation:** TextOCVP operates on object-centric slot representations, whereas MAGE and $\text{MAGE}_{\text{DINO}}$ rely on holistic, dense scene tokens (vector-quantized image latents). Whereas slots provide a factorized latent structure with one representation per object, the holistic VQ-token grid entangles ob-

---

[1] `https://github.com/Youncy-Hu/MAGE`
[2] `https://github.com/seervideodiffusion/SeerVideoLDM/tree/main`

Table 5: Comparison of MAGE variants on CLIPort. MAGE$_{\text{DINO}}$ clearly outperforms the original MAGE variant across all metrics.

| Method | PSNR ↑ | SSIM ↑ | LPIPS ↓ |
|---|---|---|---|
| MAGE | 7.116 | 0.453 | 0.713 |
| MAGE$_{\text{DINO}}$ | **23.723** | **0.940** | **0.064** |

ject information across many spatial tokens without explicit boundaries. This distinction enables a clean comparison between object-centric and holistic autoregressive models.

**Mechanism for text conditioning:** Both methods use cross-attention to incorporate language guidance, but differ in how textual information interacts with the predictor. MAGE and MAGE$_{\text{DINO}}$ compute a *single* global Motion Anchor via one cross-attention step between text and image latents, and inject this global signal uniformly into all decoding steps. In contrast, TextOCVP applies text-to-slot cross-attention within *every* transformer block, allowing the model to repeatedly integrate and select the textual information most relevant at each processing stage.

### C.2 SEER

SEER is a diffusion-based model for language-guided video prediction. It employs an Inflated 3D U-Net derived from a pretrained text-to-image 2D latent diffusion model (Rombach et al., 2022), extending it along the temporal axis and integrating temporal attention layers to simultaneously model spatial and temporal dynamics. For the language conditioning module, SEER introduces a novel Frame Sequential Text (FSText) Decomposer, which decomposes global instructions generated by the CLIP text encoder (Radford et al., 2021) into frame-specific sub-instructions. These are aligned with frames using a transformer-based temporal network and injected into the diffusion process via cross-attention layers. We initialize SEER from a checkpoint pretrained on the Something-Something V2 dataset (Goyal et al., 2017), and further fine-tune it for a few epochs. We observed that incorporating a text loss enhanced SEER's performance, while other hyper-parameters were kept consistent with its original implementation.

### C.3 Non-OC

Non-OC is a variant of our proposed TextOCVP model in which the slot-based object-centric latent representations are replaced with a single, high-dimensional slot embedding. This design allows us to isolate the contribution of object-centric structure in the latent space.

Non-OC mirrors the TextOCVP architecture, using the same visual backbone, text-guided autoregressive predictor and decoder, and it is trained with identical hyper-parameters and training schedule. The only difference lies in the scene parsing module: instead of slot attention, Non-OC applies an additional convolutional block followed by average pooling to produce a single latent vector per frame. This results in one 512-dimensional embedding, in contrast to the set of 128-dimensional object slots used in the object-centric model.

## D Datasets

### D.1 CATER

CATER (Girdhar & Ramanan, 2020) is a dataset that consists of long video sequences, each described by a textual caption. The video scenes consist of multiple 3D geometric objects in a 2D table plane, which is split into a $6 \times 6$ grid with fixed axis, allowing the exact description of object's positions using coordinates. The text instruction describes the movement of specific objects through four atomic actions: 'rotate', 'pick-place', 'slide', and 'contain'. The caption follows a template consisting of the subject, action, and an optional object or end-point coordinate, depending on the action. The movement of the objects starts at the same time

step. Furthermore, the initial positions are randomly selected from the plane grid, and the camera position is fixed for every sequence.

In our work, we employ CATER-hard, which is a complete version of the CATER dataset, containing 30000 video-caption pairs, with video frames resized to $64 \times 64$. It includes 5 possible objects: cone, cube, sphere, cylinder, or snitch, which is a special small object in metallic gold color, shaped like three intertwined tori. Furthermore, every object is described by its size (small, medium, or large), material (metal or rubber), and color (red, blue, green, yellow, gray, brown, purple, cyan, or gold if the object is the snitch), and this description is included in the textual caption. Every atomic action is available. The 'rotate' action is afforded by cubes, cylinders and the snitch, the 'contain' action is only afforded by the cones, while the other two actions are afforded by every object. Every video has between 3 and 8 objects, and two actions happen to different objects at the same time. The vocabulary size is 50.

### D.2 CLIPort

CLIPort (Shridhar et al., 2022) is a robot manipulation dataset, consisting of video-caption pairs, i.e. long videos whose motion is described by a textual video caption. There are many variants of the CLIPort dataset, but we focus on the *Put-Block-In-Bowl* variant. We generate 21000 video-caption pairs with resolution $336 \times 336$. Every video contains 6 objects on a 2D table plane, and a robot arm. Objects can be either a block or a bowl, and there is at least one of them in every sequence. The starting position of each object is random, with the only constraint being that it must be placed on the table. Each video describes the action of the robot arm picking a block, and putting it in a specific bowl. The video caption follows the template 'put the [color] block in the [color] bowl'. Each individual object in the scene has a different color. In the train and validation set, the block and the bowl that are part of the caption can have one of the following colors: blue, green, red, brown, cyan, gray, or yellow, while in the evaluation set with unseen colors they can have blue, green, red, pink, purple, white, or orange color. The other 4 objects, called distractors, can have any color. During a video sequence, it can be possible that the robot arm goes out of frame, and comes back in later frames, thus requiring the model to leverage long range dependencies. The vocabulary size is 15.

## E Evaluation Metrics

To measure TextOCVP's video prediction performance and compare it with existing approaches, we evaluate the visual quality of predicted video frames using popular image- and video-based metrics.

PSNR and SSIM (Wang et al., 2004) measure pixel-wise and statistical differences between the predicted and ground-truth video frames, respectively.

LPIPS (Zhang et al., 2018) is a perceptual metric that measures the visual similarity between two images based on deep features from pretrained neural networks, usually VGG (Simonyan & Zisserman, 2015). Unlike pixel-wise metrics, LPIPS compares activations from multiple layers and captures differences in texture, structure, and semantics, thus aligning closely with human perception.

While these metrics focus on frame-level visual quality, video prediction evaluation requires quantifying temporal consistency and motion realism. JEDi (Luo et al., 2025) measures the quality and realism of generated videos by comparing feature distributions of generated and real videos, capturing both visual fidelity and motion dynamics. We prefer JEDi over FVD (Unterthiner et al., 2019), as JEDi is less sensitive to small evaluation datasets, and more stable in low-motion synthetic scenarios.

In our evaluations and comparisons with baselines, we favor the LPIPS and JEDi scores, which correlate well with human perception, while reporting other metrics for completeness.

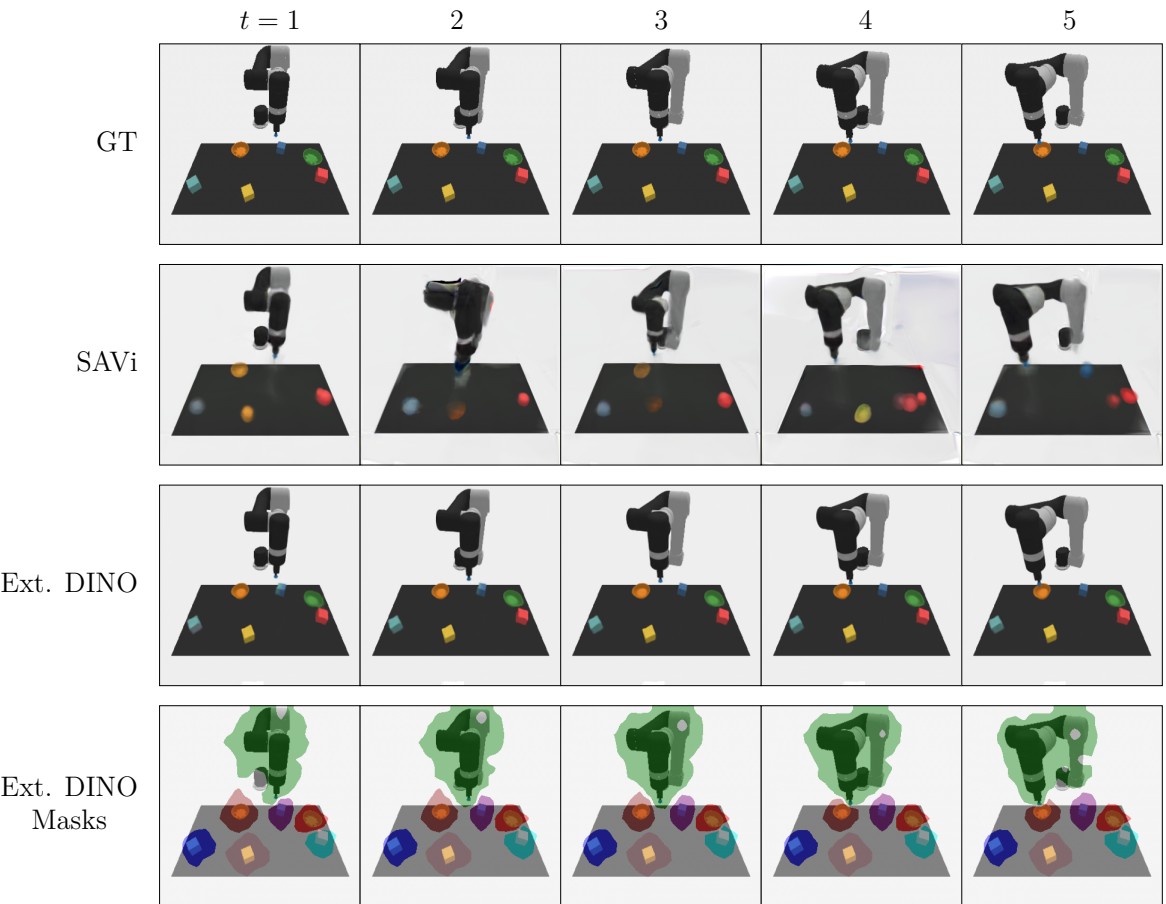

Figure 7: Comparison between SAVi and our Extended DINOSAUR (Ext. DINO) decomposition modules for reconstructing a CLIPort sequence. We visualize the reconstructed frames, as well as the slot masks obtained by Extended DINOSAUR. SAVi fails to reconstruct most objects, whereas Extended DINOSAUR accurately reconstructs the scene, while representing each object.

## F   Additional Results

### F.1   SAVi vs. DINOSAUR

Current object-centric approaches for video prediction are limited to relatively simple synthetic datasets, and struggle to scale beyond scenes featuring simple 3D shapes with simple deterministic motion. We attribute this limitation primarily to the object-centric modules used for learning object representations. Motivated by this observation, we extend the recent DINOSAUR (Seitzer et al., 2023) framework to recursively process video data and reconstruct video frames from their corresponding object-centric representations.

To demonstrate the significance of the object-centric module in scaling to more complex datasets, we compare both SAVi and our Extended DINOSAUR trained on CLIPort. As illustrated in Fig. 7, SAVi struggles to accurately represent the objects on the table, missing multiple objects and changing their shape and color. In contrast, the Extended DINOSAUR model successfully reconstructs the scene, closely resembling the input, while accurately representing each object. The visual features extracted by the DINOv2 (Oquab et al., 2024) encoder contain high-level semantic information, and during training, the slots are specifically optimized to efficiently encode this information. This design enables the model to scale and handle more complex object-centric video data effectively.

Table 6: Average inference (Inf.) time for $T = 1$ and $T = 9$ frame predictions on CATER and CLIPort.

| | CATER Inf. [ms] | | CLIPort Inf. [ms] | |
|---|---|---|---|---|
| Model | $T = 1$ | $T = 9$ | $T = 1$ | $T = 9$ |
| MAGE | $15.4 \pm 1$ | $111.1 \pm 11$ | $34.4 \pm 3$ | $198.8 \pm 2$ |
| TextOCVP | $\mathbf{13.6 \pm 1}$ | $\mathbf{78.6 \pm 7}$ | $\mathbf{21.1 \pm 1}$ | $\mathbf{109.9 \pm 2}$ |

Table 7: Evaluation at prediction horizons $T = 9$ and 19. TextOCVP is the best performing among all compared methods both on CATER and CLIPort, followed by MAGE/MAGE$_{\text{DINO}}$. Best two results are shown in bold and underlined, respectively.

| | CATER$_{1\to9}$ | | | | CATER$_{1\to19}$ | | | | CLIPort$_{1\to9}$ | | | | CLIPort$_{1\to19}$ | | | |
|---|---|---|---|---|---|---|---|---|---|---|---|---|---|---|---|---|
| Method | PSNR↑ | SSIM↑ | LPIPS↓ | JEDi↓ | PSNR↑ | SSIM↑ | LPIPS↓ | JEDi↓ | PSNR↑ | SSIM↑ | LPIPS↓ | JEDi↓ | PSNR↑ | SSIM↑ | LPIPS↓ | JEDi↓ |
| OCVP | 29.08 | 0.874 | 0.078 | 4.16 | 28.11 | 0.854 | 0.101 | 8.08 | – | – | – | – | – | – | – | – |
| Non-OC | 29.68 | 0.874 | 0.092 | 3.04 | 28.39 | 0.849 | 0.112 | 8.62 | 23.44 | 0.901 | 0.184 | 8.13 | 20.14 | 0.872 | 0.210 | 13.23 |
| SEER | 22.05 | 0.723 | 0.245 | 11.23 | 16.05 | 0.535 | 0.299 | 17.29 | 21.01 | 0.887 | 0.141 | 6.80 | 11.30 | 0.622 | 0.331 | 8.29 |
| MAGE | **34.91** | 0.877 | 0.108 | 3.46 | **34.76** | 0.871 | 0.111 | 5.88 | 23.72 | 0.940 | 0.064 | 2.11 | 22.27 | **0.931** | **0.075** | 2.59 |
| TextOCVP | 32.98 | **0.922** | **0.036** | **2.16** | 31.29 | **0.902** | **0.044** | **5.09** | **26.99** | **0.950** | **0.062** | **1.36** | **23.88** | **0.931** | 0.078 | **2.23** |

## F.2 Computational Efficiency

In Table 6 we report the average inference time for $T = 1$ and $T = 9$ frame predictions on CATER and CLIPort using a NVIDIA-A6000-48Gb GPU.

During inference, TextOCVP is significantly faster than MAGE/MAGE$_{\text{DINO}}$, achieving $\approx 40\%$ lower latency. This efficiency stems from our model's object-centric design, which operates on a small number of object slots, in contrast to the larger number of spatial tokens employed by MAGE.

## F.3 Quantitative Comparison

In the main paper, we present quantitative evaluations on the CATER and CLIPort datasets using three evaluation metrics, shown in two separate tables. For completeness, we report in Table 7 a text-guided video prediction evaluation of TextOCVP and multiple baseline models on CATER and CLIPort using four distinct evaluation metrics for different prediction horizons.

Our proposed TextOCVP outperforms all baselines on both the CATER and CLIPort datasets, surpassing the next-best method, MAGE/MAGE$_{\text{DINO}}$, by a clear margin. Most notably, TextOCVP consistently achieves the best LPIPS and JEDi scores, demonstrating superior frame-wise visual quality, as well as video-level fidelity and motion realism.

## F.4 Object-Centric Evaluation

A key advantage of object-centric approaches in video prediction is their ability to generate segmentation masks alongside frame predictions. For TextOCVP, we derive predicted segmentation masks by applying an `argmax` operation over all object slot masks produced by the decoder during prediction and subsequently filtering the resulting masks by assigning values with a small magnitude to the background. These predicted masks are evaluated against ground-truth segmentation masks using the Intersection over Union (IoU) metric.

We compare our model against two baselines on CLIPort. The *Copy-Seed* baseline simply replicates the predicted segmentation mask from the seed frame across all frames, while the *OC-module-only* variant generates segmentation masks by feeding ground-truth frames directly into the object-centric module, i.e., Extended DINOSAUR. As shown in Table 8, TextOCVP outperforms both baselines across prediction horizons of $T = 9$ and $T = 19$ frames, demonstrating the model's ability to model object dynamics, maintaining spatial consistency and object coherence over time. The predicted masks of TextOCVP align more closely with the

Table 8: Object-centric evaluation on CLIPort. Comparison of Intersection over Union (IoU) scores between predicted and ground-truth segmentation masks for different prediction horizons ($T = 9$ and $T = 19$).

| Method | $\textbf{CLIPort}_{1 \rightarrow 9}$ IoU↑ | $\textbf{CLIPort}_{1 \rightarrow 19}$ IoU↑ |
|---|---|---|
| Copy-Seed | 0.540 | 0.525 |
| OC-module-only | 0.554 | 0.553 |
| TextOCVP | **0.573** | **0.569** |

Table 9: Quantitative evaluation of TextOCVP's controllability on CLIPort. We report the mean pick-and-place success rates (averaged over 5 runs) for the original evaluation set and three instruction variants differing in the target bowl. TextOCVP maintains consistent performance across variants, reflecting strong robustness to instruction changes.

| Method | Pick-and-Place success rate | | | | |
|---|---|---|---|---|---|
| | **Original set** | **Variant 1** | **Variant 2** | **Variant 3** | **Mean variants** |
| TextOCVP | 0.83 | 0.86 | 0.80 | 0.81 | 0.82 |

ground-truth segmentations, benefiting from better temporal stability throughout the prediction horizon. In Figs. 11 and 12 , we illustrate TextOCVP's object-centric behavior on CATER and CLIPort by showing the predicted segmentation masks, as well as different per-object predictions.

### F.5 Quantitative Evaluation of Controllability

In Sec. 4.3.4, we qualitatively show how TextOCVP is able to adapt its predictions based on the language instruction it receives as input. We further make an initial attempt to quantitatively evaluate the controllability of TextOCVP.

Starting from a CLIPort evaluation set of 100 sequences, we create three language instruction variants for each sequence, differing in the specified target bowl. Given the same initial frame, TextOCVP then generates future predictions conditioned on the adapted instructions. To assess performance, we ground the slot masks produced by the video rendering decoder to their corresponding objects in the scene and compute two key distances: (1) between the masks of the robot arm and the specified block, and (2) between the picked block and the target bowl. These measures are used to estimate the pick-and-place success rate. A generated sequence is considered successful only if the robot arm remains sufficiently close to the correct block over multiple frames and the block is significantly close to the target bowl toward the end of the sequence.

As shown in Table 9, TextOCVP demonstrates consistent success rates across different instruction variations, indicating strong robustness and fine-grained controllability. Given identical starting scene, the model effectively adapts its predictions to the changing text instructions, successfully placing the block into the specified bowls in most of the cases.

We note that an equivalent experiment could not be performed with MAGE$_{\text{DINO}}$, as it does not generate object masks during prediction.

### F.6 Robustness to Number of Objects

In Sec. 4.3.2, we quantitatively assess the generalization and robustness of TextOCVP in video prediction tasks involving novel scene compositions. Our results highlight the benefits of object-centric representations over holistic scene-based approaches.

This finding is further illustrated in Fig. 9, which presents qualitative comparisons of video generations for scenes containing eight objects, in contrast to the six-object configurations seen during training. As

Table 10: Impact of visual artifacts on TextOCVP's performance on CLIPort. Cropping the artifact-prone bottom part of the predicted frames leads to a notable improvement in TextOCVP's performance.

| | **CLIPort$_{1 \to 9}$** | | **CLIPort$_{1 \to 19}$** | |
| Image View | SSIM↑ | LPIPS↓ | SSIM↑ | LPIPS↓ |
|---|---|---|---|---|
| Full Image | 0.950 | 0.062 | 0.931 | 0.078 |
| Excluded Bottom | **0.953** | **0.050** | **0.932** | **0.069** |

observed, our model correctly predicts sequences following the motion described in the text instructions, whereas MAGE$_{\text{DINO}}$ fails to generate accurate sequences according to the descriptions.

These results further demonstrate the effectiveness of object-centric representations for video prediction, as TextOCVP is able to generalize to scenes with more objects by simply increasing the number of slots. This flexibility is enabled by initializing object slots via sampling from a learned Gaussian distribution, allowing the use of a variable number of slots at test time while breaking symmetry and preserving permutation invariance. Although the object slots do not specialize during training, they reliably bind to meaningful entities through iterative attention during inference. This results in robust and scalable scene decomposition, enabling accurate modeling of complex scenes with varying object counts.

### F.7 Impact of Visual Artifacts

As already discussed in the main paper, TextOCVP occasionally generates visual artifacts on the CLIPort dataset, most noticeably in the bottom-center region of the frame, where blurry patches often appear.

To assess the impact of these visual artifacts on TextOCVP's quantitative performance, we evaluate our model after removing the last bottom rows from the predicted frames—an area that contains only background pixels and the artifacts. As shown in Table 10, removing these rows leads to improved results, particularly for the perceptual LPIPS metric. These findings verify that TextOCVP generates future frame predictions that closely follow the text description, and that its overall performance is underestimated due to localized visual artifacts.

### F.8 Impact of Video Rendering Module

In the previous section (Appendix F.7), we described how visual artifacts affect TextOCVP's performance on the CLIPort dataset. We argue that these artifacts mostly originate from limitations in the video rendering module, and that they can be mitigated by adopting a more expressive decoder architecture.

To support this claim, we conducted an additional experiment on CLIPort in which we replaced the simple CNN decoder—described in Appendix B.1—with a more expressive alternative, while keeping the rest of TextOCVP's architecture unchanged. The new decoder, inspired by VQGAN (Esser et al., 2021), integrates convolutional, residual, non-local attention, and upsampling blocks, and contains nearly twice as many learnable parameters as the original module ($\sim 25$M vs. $\sim 13$M).

TextOCVP models using both decoder variants are qualitative compared in Fig. 10, which shows that the more expressive decoder substantially reduces the visual artifacts observed in the original model's predictions. In Table 11 we quantitatively compare both decoder variants. The more expressive decoder model achieves noticeably better LPIPS and JEDi scores, demonstrating superior perceptual frame and video quality.

Overall, these findings indicate that employing a more capable decoder can mitigate some of TextOCVP's visual limitations. They also highlight the benefits of our modular architecture, which enables such improvements to be incorporated seamlessly without modifying the underlying predictor or representation. Exploring more powerful rendering modules—such as diffusion-based decoders (Jiang et al., 2023) or refining the VQGAN-style architecture (Esser et al., 2021)—is a promising direction for future work.

Table 11: Comparison of TextOCVP with a variant using a more expressive video rendering module on CLIPort. The stronger, VQGAN-inspired (Esser et al., 2021), decoder leads to improved perceptual quality.

| TextOCVP Decoder | CLIPort$_{1 \to 9}$ | | CLIPort$_{1 \to 19}$ | |
| --- | --- | --- | --- | --- |
| | LPIPS↓ | JEDi↓ | LPIPS↓ | JEDi↓ |
| Simple CNN | 0.062 | 1.36 | 0.078 | 2.23 |
| VQGAN-based | **0.043** | **0.86** | **0.065** | **2.22** |

(a) LPIPS (↓) metric across prediction horizons      (b) IoU (↑) metric across prediction horizons

Figure 8: LPIPS and slot-mask IoU metrics across prediction horizon of 40 future frames. The plots show both the average values (bold line) as well as the standard deviation (shaded areas).

### F.9 Evaluation of Predictions over Time

In Fig. 8, we present plots of LPIPS and slot-mask IoU scores over prediction horizons of up to 40 future frames. These curves show that our model's predictions steadily deviate from the ground truth as the prediction horizon increases, reflecting the natural accumulation of errors inherent to autoregressive prediction.

We also note that quantitative metrics such as LPIPS or IoU become less informative when predicting beyond 20-30 future frames. At these prediction horizons, even slight differences in object velocity or trajectory can compound and lead to large discrepancies in frame-wise metrics, despite still producing qualitatively consistent futures. This behavior is typical of autoregressive video models and reflects the sensitivity of pixel-level scores to small temporal deviations, rather than a qualitative failure of the predicted dynamics.

### F.10 Additional Qualitative Evaluations

Fig. 13 shows an example where we evaluate TextOCVP and MAGE$_{DINO}$ for text-guided video prediction over a long prediction horizon of 50 frames. MAGE$_{DINO}$ fails to complete the task outlined in the textual description, as it stops generating consistent robot motion after 30 frames. In contrast, TextOCVP successfully predicts future frames where the robot completes the pick-and-place task.

Figs. 14 and 15  show qualitative evaluations on CATER in which both MAGE and TextOCVP successfully predict sequence continuations following the instructions from the textual description.

Fig. 16 illustrates an example where MAGE fails to generate a correct sequence, while TextOCVP successfully completes the task described by the text.

Figs. 17 and 18  show examples of TextOCVP's control over the predictions. In both sequences, TextOCVP generates a correct sequence given the text instructions, and seamlessly adapts its generations to a modified version of the textual instructions.

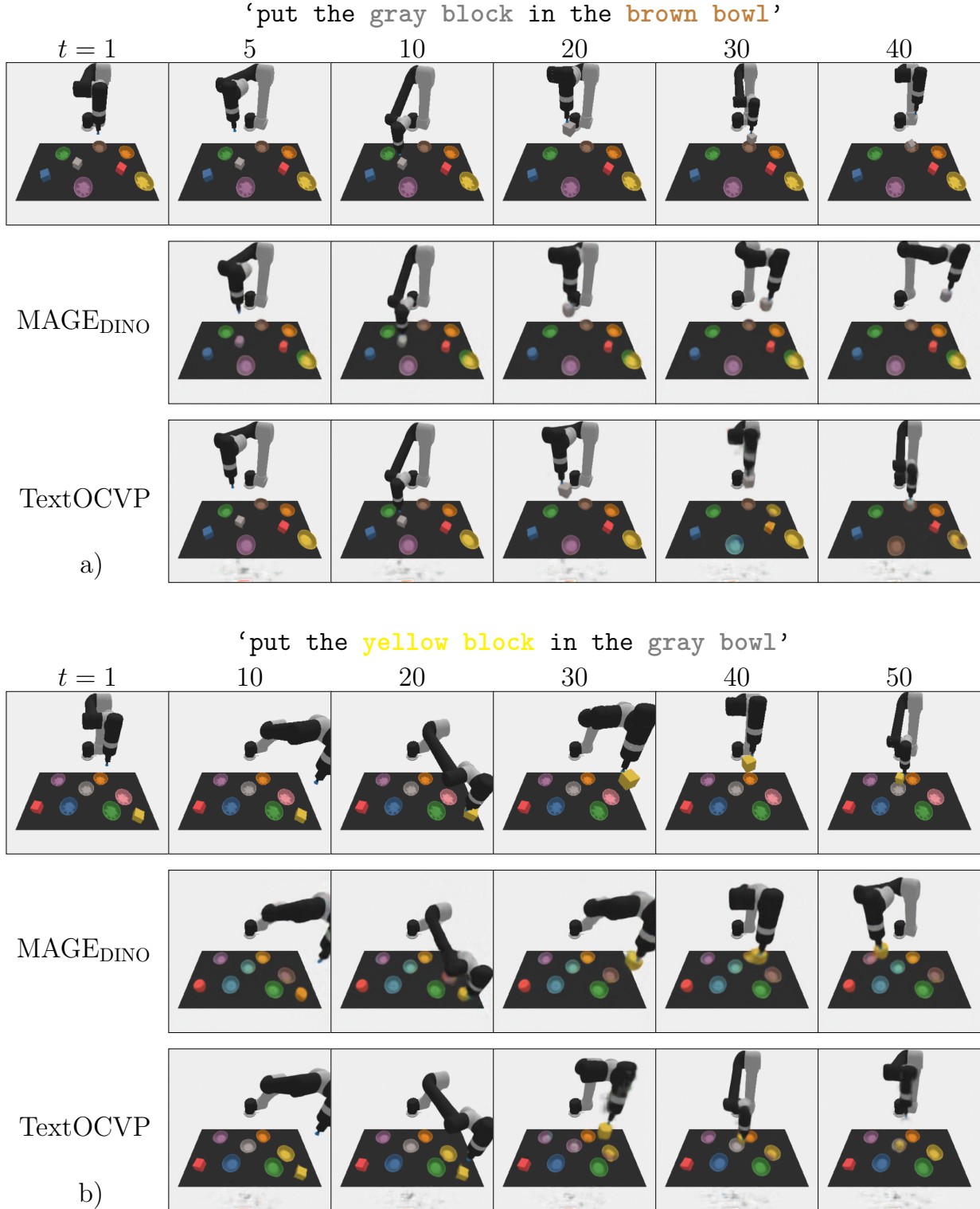

Figure 9: Qualitative evaluation of MAGE_DINO and TextOCVP on CLIPort sequences with more objects than those seen during training (eight instead of six). TextOCVP correctly generates sequences according to the text instructions, whereas MAGE_DINO misses the target bowl.

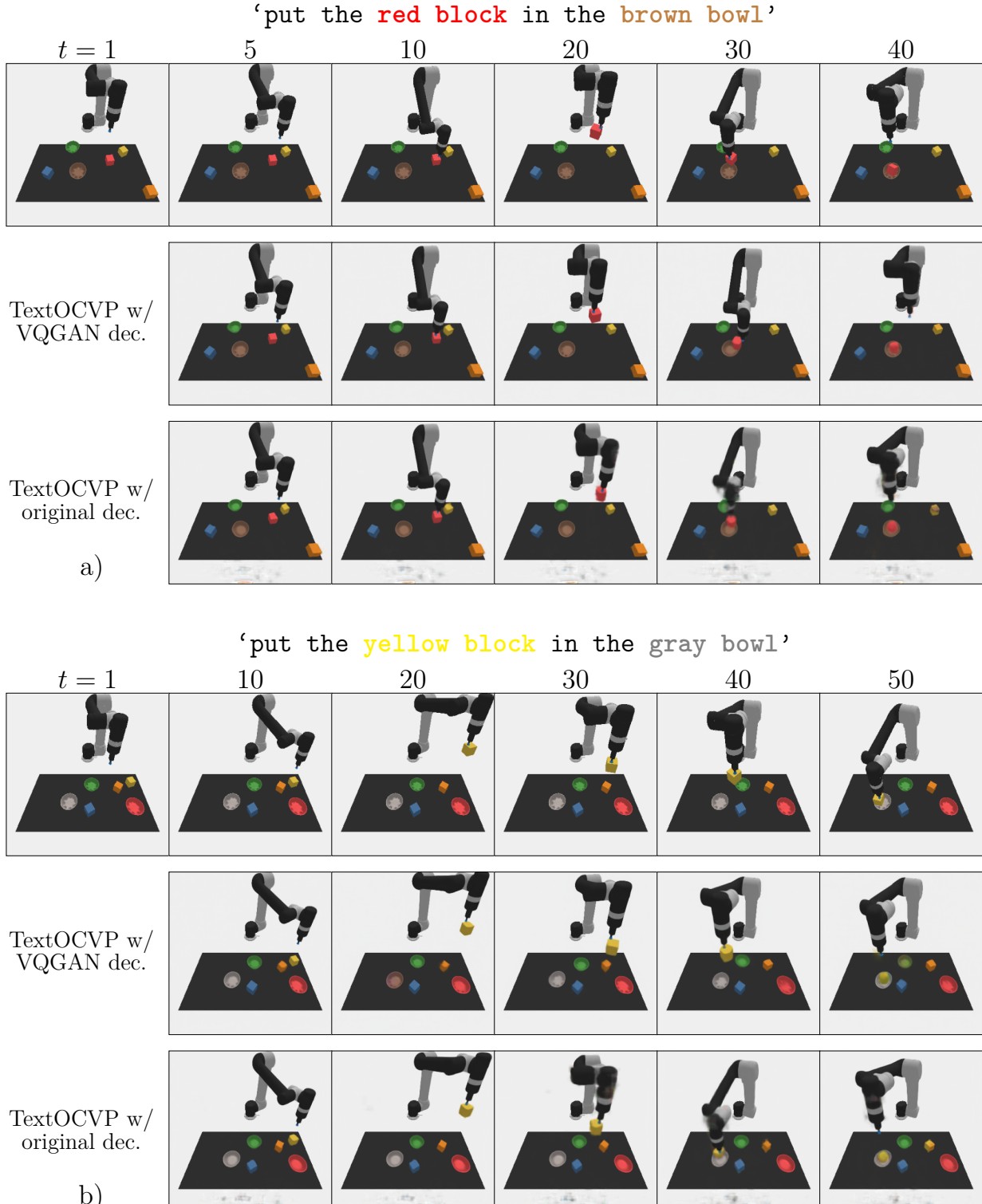

Figure 10: Qualitative comparison between original TextOCVP implementation and a TextOCVP variant with a more expressive, VQGAN-inspired, video rendering module. While both variants correctly predict the described motion, the new decoder improves the predicted frames' quality by mitigating most of visual artifacts, especially on the bottom-center region and in the robot arm.

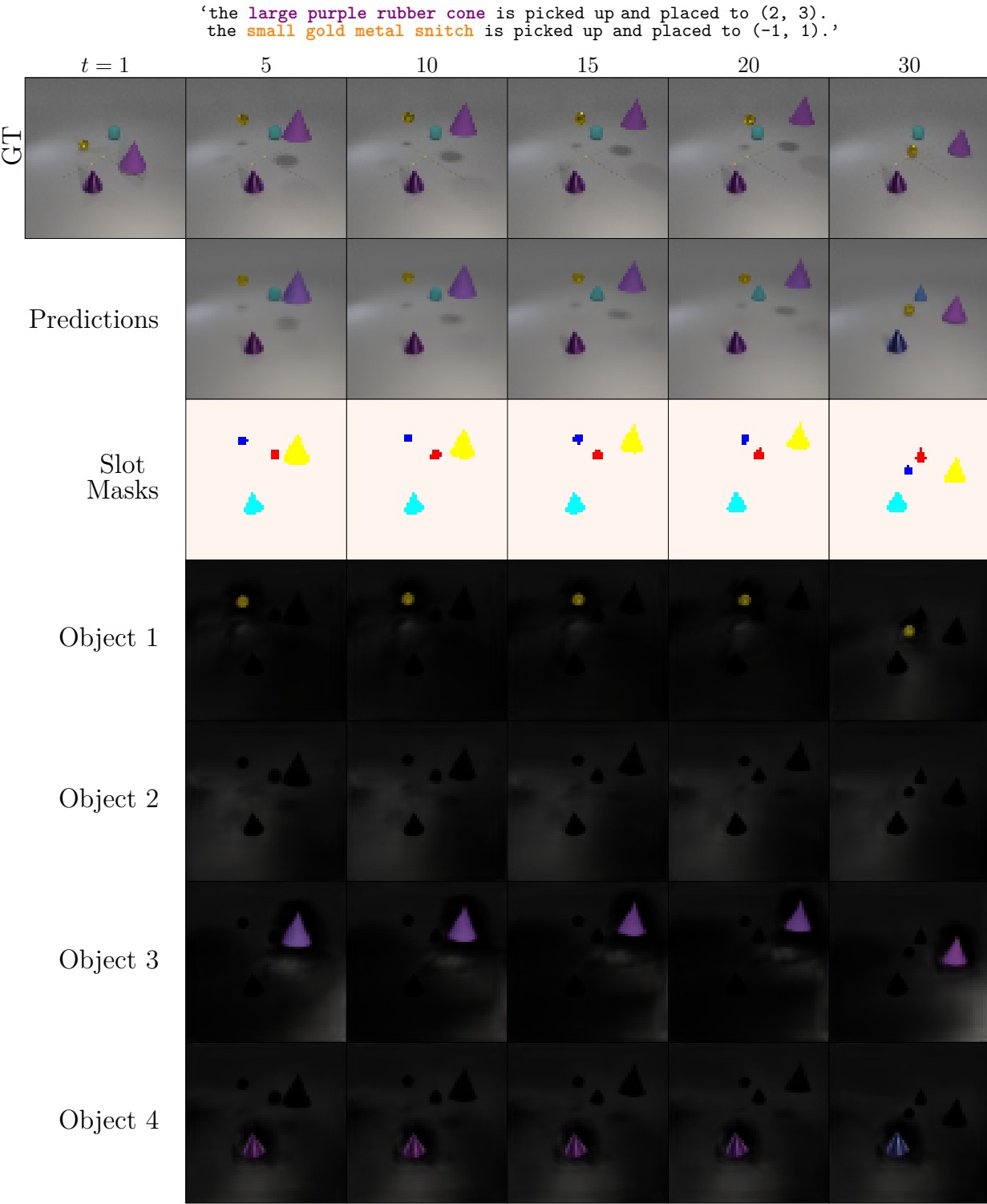

Figure 11: TextOCVP's object-centric behavior on a CATER sequence. The first row shows the ground truth sequence, followed by TextOCVP's predicted frames and segmentation masks. The subsequent rows display the reconstructed objects from four of the predicted slots across various time steps, highlighting the ability of TextOCVP to model the dynamics of individual objects in the scene through slot representations.

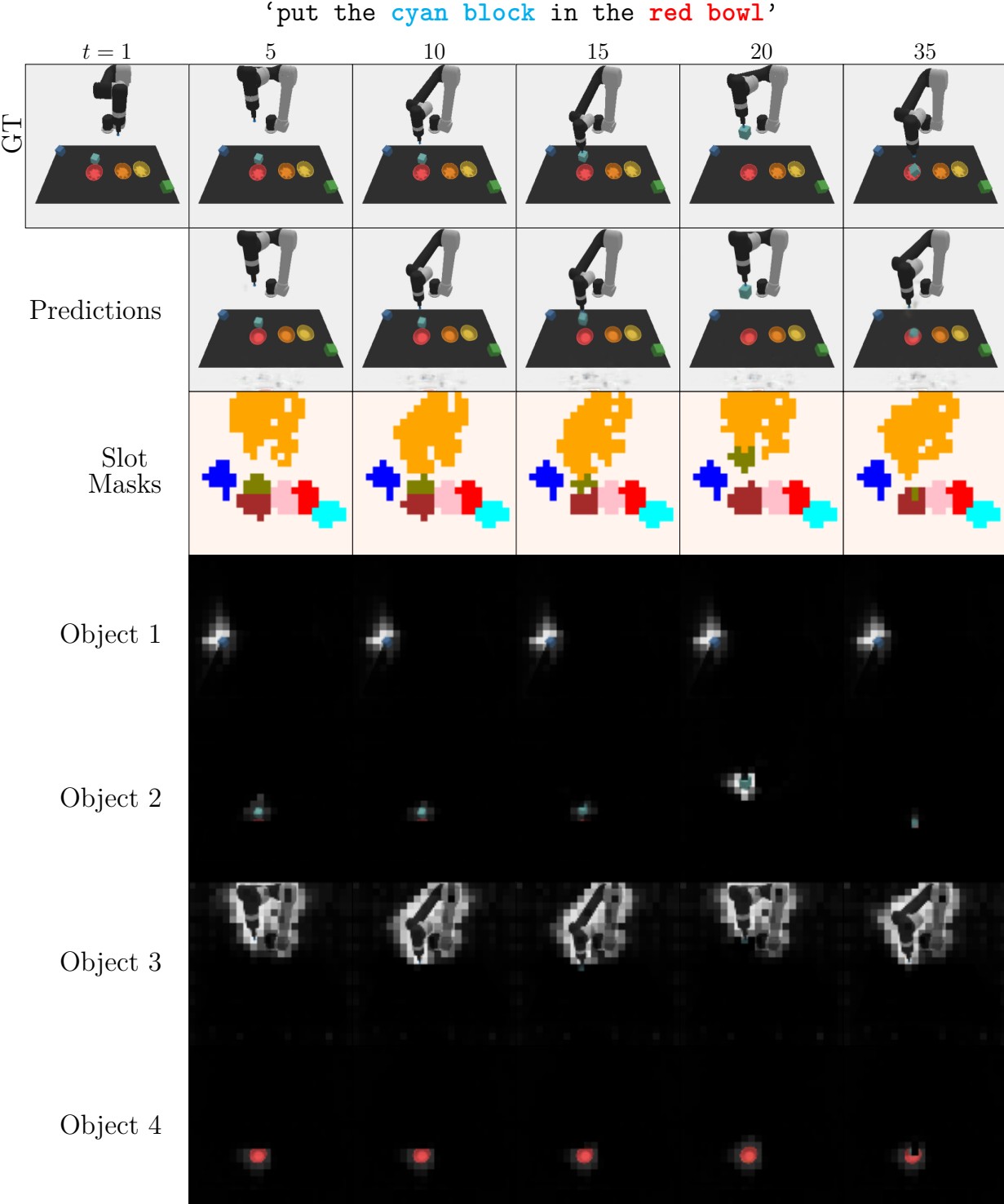

Figure 12: TextOCVP's object-centric behavior on a CLIPort sequence. The first row shows the ground truth sequence, followed by TextOCVP's predicted frames and segmentation masks. The subsequent rows illustrate the represented objects from four of the predicted slots across various time steps. Although only eight slots are required for this dataset (six objects, one robot arm, and one background), we use ten slots in CLIPort experiments which proved to beneficial; the two extra slots represent background. We emphasize that the object segmentations are computed at the patch level, thus the pixelated appearance in the visualizations.

'put the gray block in the brown bowl'

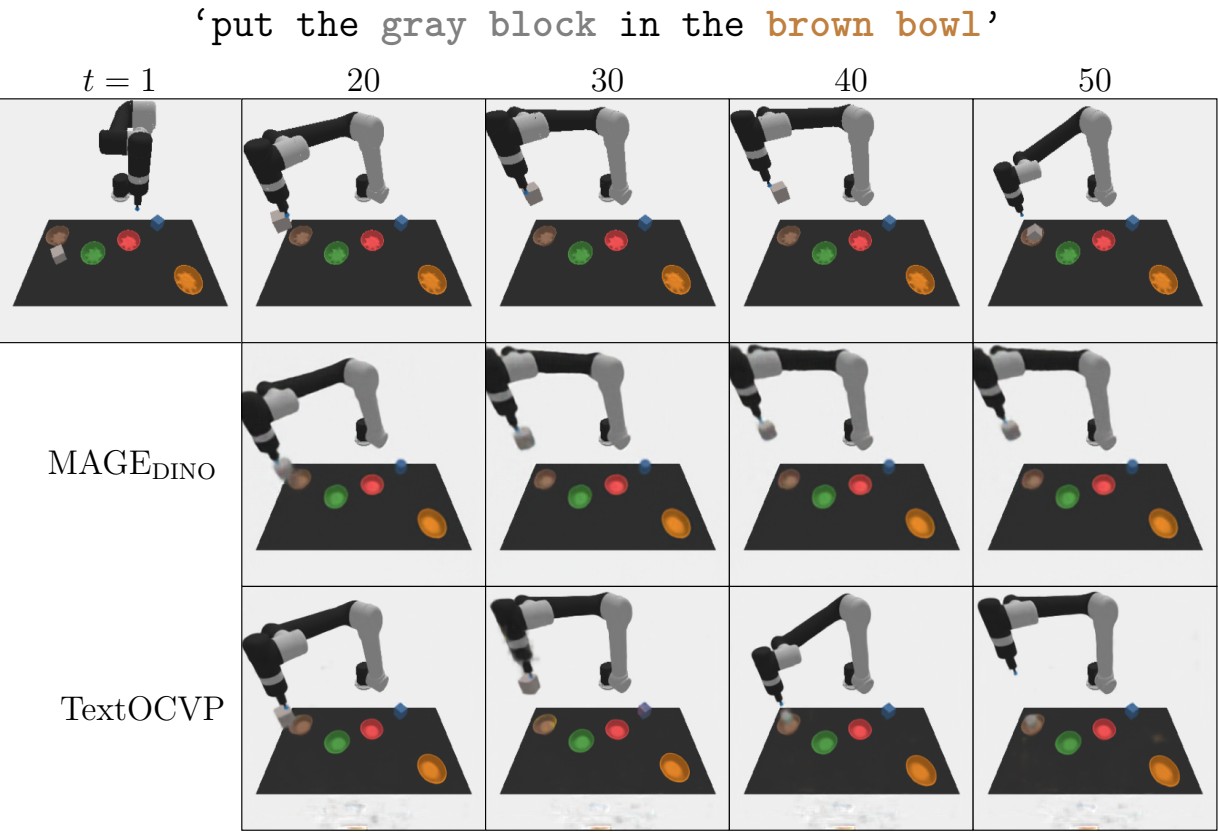

Figure 13: Qualitative result on CLIPort. Top row shows ground truth frames. TextOCVP completes the pick-and-place task, whereas MAGE_DINO fails to predict the robot motion.

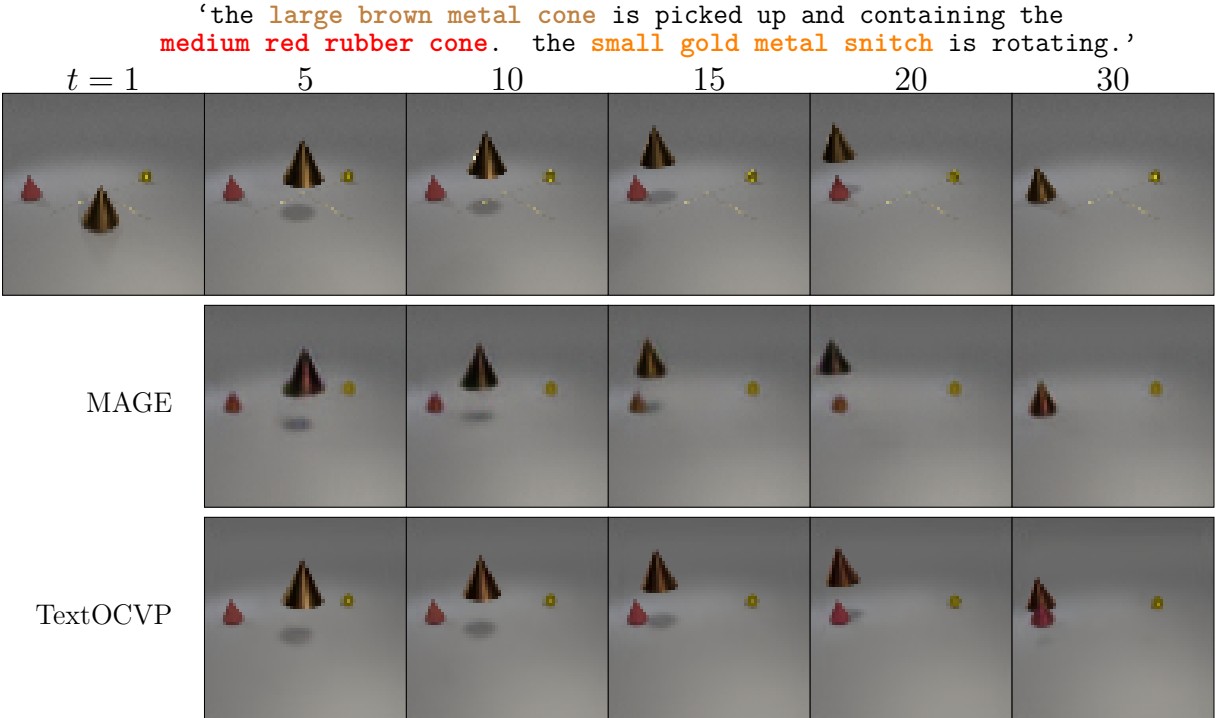

Figure 14: Qualitative evaluation on CATER. Both MAGE and TextOCVP successfully generate a sequence following the instructions from the textual description.

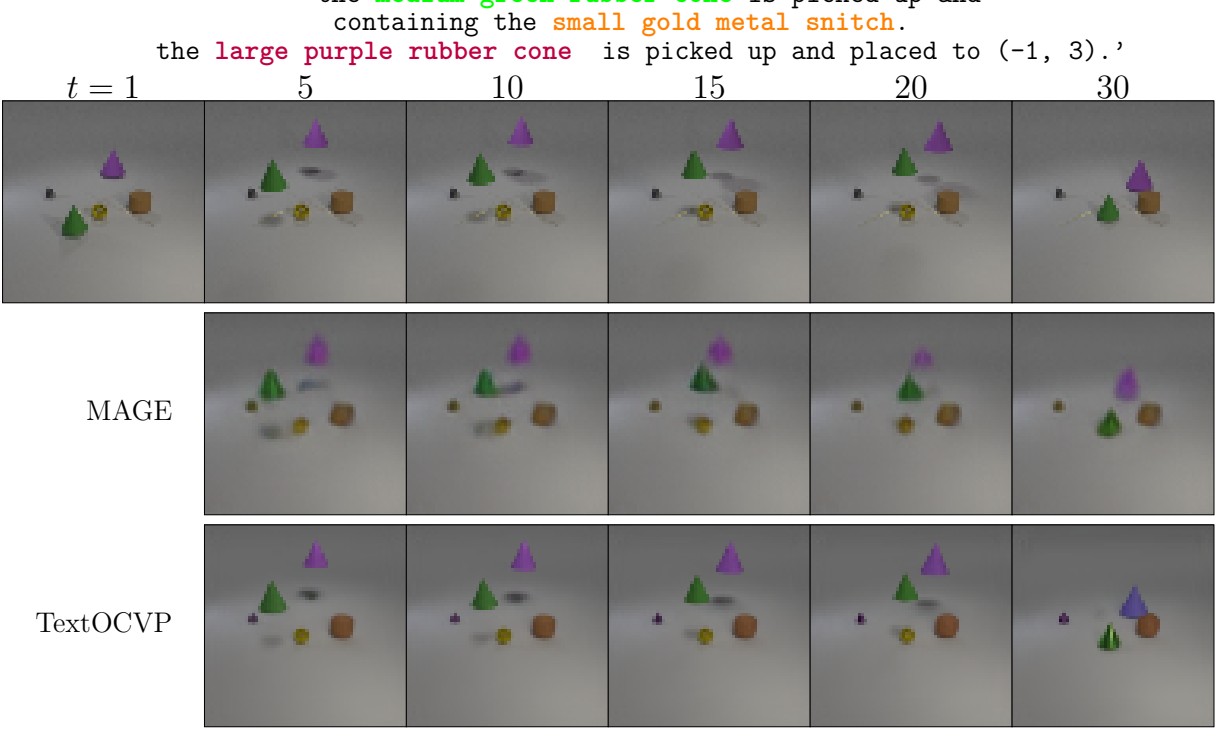

Figure 15: Qualitative evaluation on CATER. Both MAGE and TextOCVP successfully generate a sequence that illustrates the motion described in the text, but MAGE's predictions are of a lower resolution.

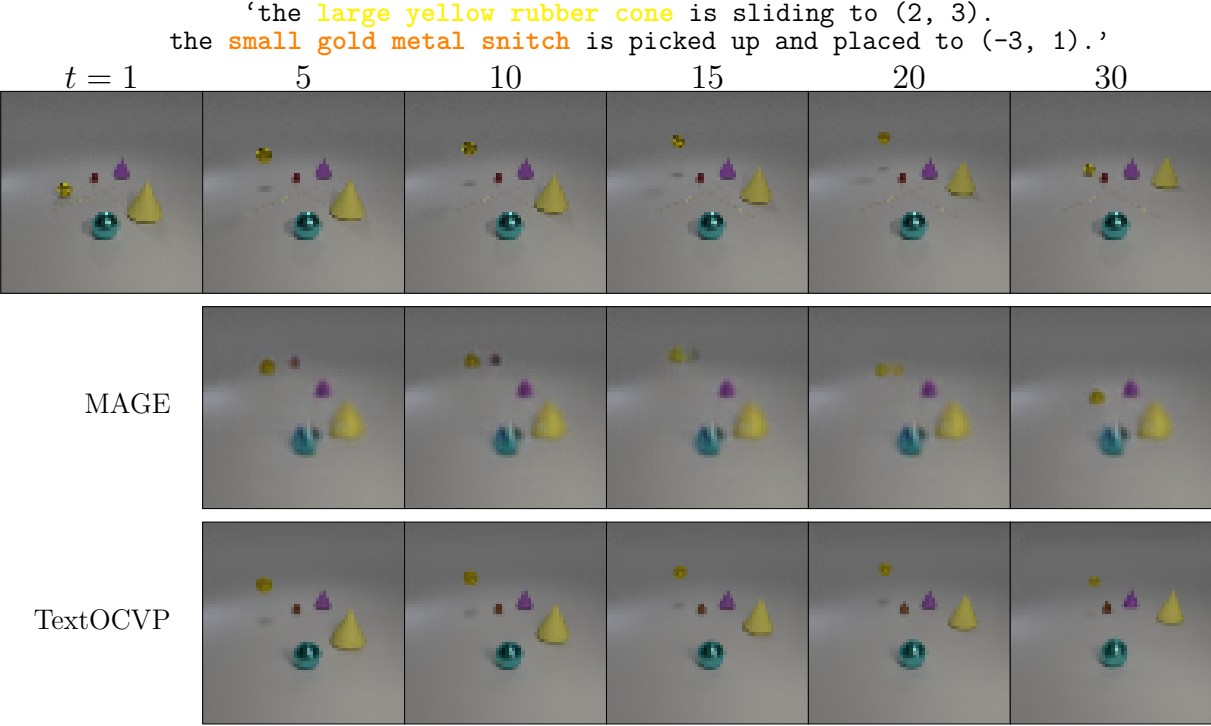

Figure 16: Qualitative evaluation on CATER. MAGE fails to generate a sequence that accurately follows the motion described in the text. Specifically, the yellow cone does not slide as expected, and artifacts such as the merging of two small objects are introduced. On the other hand, the sequence generated by TextOCVP is closely aligned with the ground truth, accurately capturing the motion of the objects.

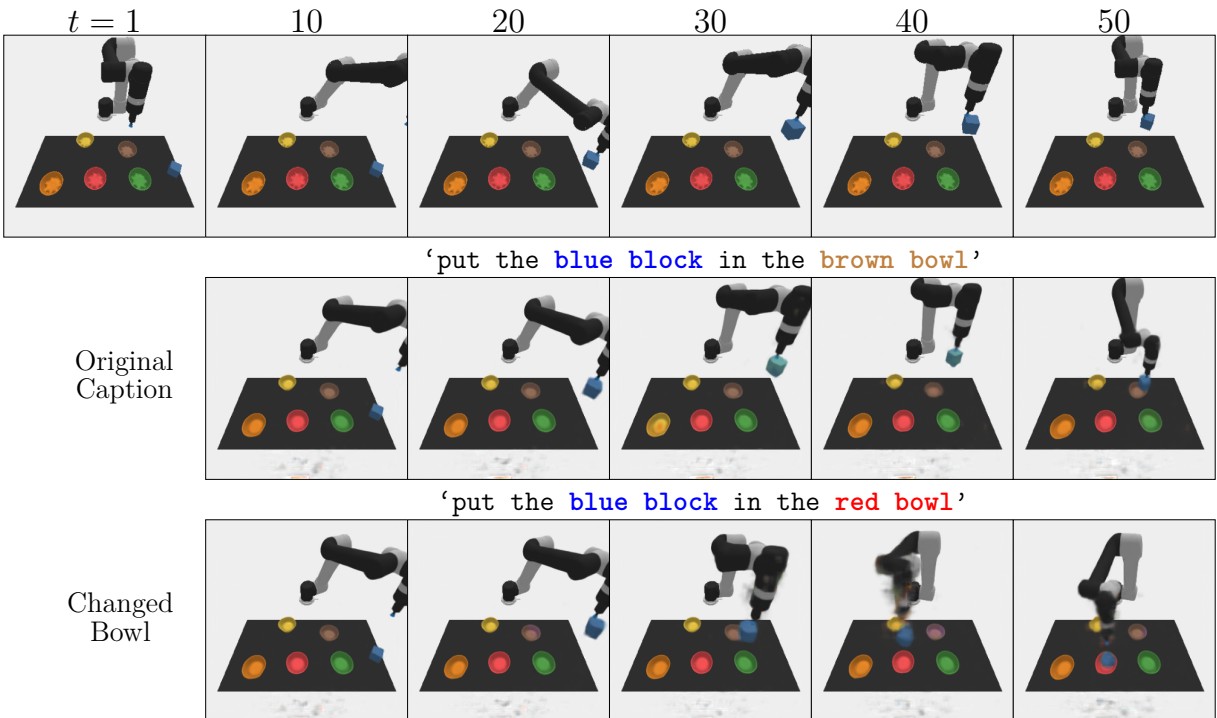

Figure 17: Qualitative evaluation of TextOCVP controllability on CLIPort. TextOCVP correctly generates a sequence where the robot picks up and places the block specified in the textual instruction.

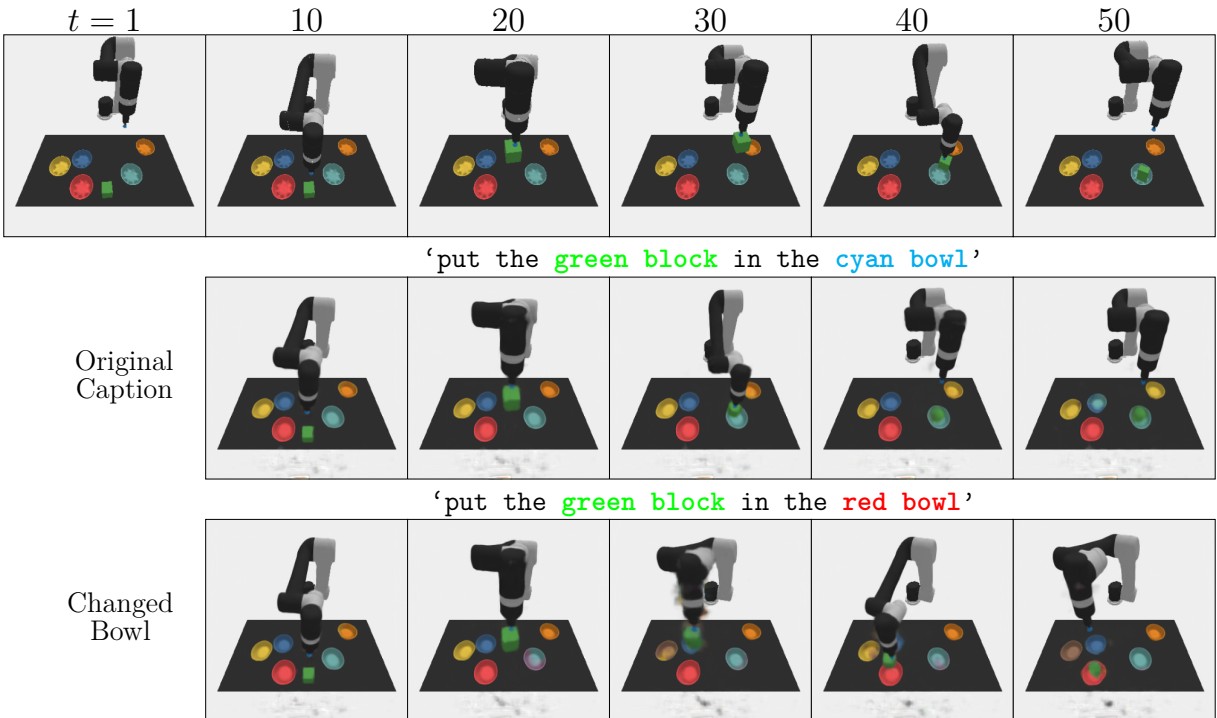

Figure 18: Qualitative evaluation of TextOCVP controllability on CLIPort. TextOCVP correctly generates a sequence where the robot picks up and places the block specified in the textual instruction.

