# OpenReview forum: "TextOCVP: Object-Centric Video Prediction with Language Guidance"
_TMLR — Accepted by TMLR_

### Review · Reviewer_q6ZE · 2025-11-19

**Summary Of Contributions:**

The paper proposes a video prediction model that leverages object-centric representations conditioned on natural language instructions.

Contributions:
- A modular framework that parses frames into object slots and predicts future states using a text-conditioned transformer.
- Empirical demonstration that object-centric representations significantly outperform holistic baselines in out-of-distribution scenarios, specifically regarding unseen object colors and higher object count.
- Quantitative and qualitative evidence that the model can precisely follow fine-grained text instructions for robotic manipulation tasks.

Strengths:
- The integration of text-to-slot attention is well-motivated and provides interpretability.
- The contribution for handling complex textures in CLIPort is valuable.
- The paper provides strong empirical support for the long-held hypothesis that object-centric biases facilitate combinatorial generalization.

Weaknesses:
- Evaluation is limited to synthetic and simulation datasets, despite claims regarding applicability to "robotics".
- The method suffers from visual artifacts and temporal instability at longer horizons (T>30).

**Audience:**

Yes

**Audience Explanation:**

The paper is relevant to researchers in computer vision, robot learning, and representation learning. Specifically, the Object-Centric Learning community will be interested in the method for conditioning slots on text.

**Broader Impact Concerns:**

None. The paper focuses on a fundamental method for video prediction in simulated environments. The current work does not raise ethical concerns requiring a specialized statement beyond standard considerations.

**Claims And Evidence:**

Yes

**Claims Explanation:**

The authors provide rigorous comparisons against strong baselines. The claims regarding robustness are well-supported by Table 3. The claims regarding controllability are supported by both qualitative visualizations in Fig. 6 and the ablation studies in Table 9.

While the visual fidelity is not photorealistic (as acknowledged in the limitations), the evidence supports the core claims about control and generalization dynamics.

**Requested Changes:**

I recommend the following adjustments to strengthen the submission. None are strictly critical for acceptance, but addressing them would improve the paper's precision and impact.

- The introduction claim the method addresses limitations limiting applicability in "robotics" and enables "world models for robotics". However, the experiments use CLIPort, which is a simulation environment. Please soften the language to specify "robotic simulation" or "simulated manipulation environments" to avoid overclaiming real-world applicability, or explicitly discuss the domain gap challenges in the limitation section.
- The paper notes that artifacts likely stem from the video rendering module. A brief discussion (or a small experiment if feasible) comparing the current decoder with a stronger alternative would be valuable.
- The limitations section mentions jittering after 30 frames. It would be helpful to see a plot of prediction error (e.g., LPIPS or slot position error) over time (T=1 to 50) in the appendix.
- For the SEER baseline, ensure the training compute/data regime is explicitly compared. Diffusion models often require more data. Clarifying if SEER was under-trained or if this represents a fundamental efficiency advantage of TextOCVP would strengthen the results analysis.

---

> ### Author Response · Authors · 2025-11-28
> **Author response to Reviewer q6ZE**
>
> We thank the reviewer for their valuable feedback. Below we address all concerns and describe the changes made in the revised manuscript.
>
> ---
>
> **Refining certain claims in the paper**
>
> We agree with the reviewer and have now softened the language throughout the manuscript. In particular, we now refer to “robotic simulations” or “simulated robotic environments” rather than “robotics”, to avoid overclaiming and to more accurately reflect the scope of our evaluation.
>
> ---
>
>
> **Addressing visual artifacts: comparison with a stronger decoder**
>
> To address the reviewer’s suggestion, we conducted an additional experiment on CLIPort where we compare our original TextOCVP architecture with a variant using a more expressive, VQGAN-inspired, video rendering module.
>
> This experiment, reported in Appendix F.8, shows that the more expressive decoder can alleviate some of the visual limitations of TextOCVP, improving upon the quantitative evaluation while also reducing the observed visual artifacts.
> Exploring more powerful rendering modules—such as diffusion-based decoders or refining the VQGAN-style architecture—is a promising direction for future work.
>
>
> ---
>
>
> **Regarding temporal instability beyond 30 frames**
>
> We have added in Appendix F.9 plots depicting the LPIPS and slot-mask IoU over prediction horizons of up to 40 future frames.
> These curves show that our model’s predictions steadily deviate from the ground truth as the prediction horizon increases, reflecting the natural accumulation of errors inherent to autoregressive prediction. They also highlight how, beyond roughly 20 frames, pixel-level and perceptual metrics become increasingly sensitive to minor temporal misalignments, even when the predicted sequences remain qualitatively coherent.
>
> ---
>
> **Clarifications regarding SEER training**
>
> In our experiments, we initialize SEER with the publicly available SSv2 checkpoint and fine-tune it for 80k iterations on a single A6000 GPU, closely following the training recipe and hyper-parameters recommended in the original implementation [1]. We empirically observed that training beyond 80k iterations did not improve the model performance. Additionally, we also experimented with training the model from scratch, but the limited amount of video data in CLIPort and CATER led to worse prediction performance.
>
> Finally, we would like to emphasize that our results are consistent with prior observations [2], which also showed limitations of SEER on structured, small/medium synthetic robotic datasets.
>
> [1] https://github.com/seervideodiffusion/SeerVideoLDM
>
> [2] Jeong et al. Language-Guided Object-Centric World Models for Predictive Control. ArXiv Preprint. 2025.
>
> ---
>
> **Regarding evaluations limited to synthetic and simulation datasets**
>
> We appreciate the reviewer's interest in extending our work to real-world settings. Our current work was intended as an initial exploration into language-conditioned object-centric video prediction, emphasizing controlled, robust, and interpretable evaluations across synthetic but diverse and well-structured environments (CATER,  CLIPort). These datasets allow us to rigorously analyze controllability and object grounding—key aspects of TextOCVP.
>
> We agree that real-world generalization is a necessary next step. Our framework is modular and scalable, and we envision future extensions that integrate powerful vision-language encoders and more expressive rendering modules. We discuss such extensions in Appendix A, and plan to pursue them in future work.

---

### Review · Reviewer_AP1H · 2025-11-22

**Summary Of Contributions:**

This paper presents TextOCVP, an object-centric model for text-guided video prediction. The method parses a reference frame into slot-based object representations and uses a text-conditioned autoregressive transformer to predict future object dynamics and render video frames consistent with a natural-language instruction. The authors evaluate two variants (based on SAVi and DINOSAUR), benchmark against existing text-conditioned video prediction methods (MAGE, SEER, MAGEDINO), and conduct ablations on model depth, text encoders, and slot count. They further analyse robustness to distribution shifts and visualise text-to-slot attention to argue interpretability and controllability.

**Key strengths**
* The paper is clearly written, well structured, and easy to follow.
* The overall idea (slot-level conditioning with language) is compelling and practically useful, particularly for robotics.
* Experiments on CATER and CLIPort are extensive and show meaningful improvements over baselines.
* The method provides fine-grained control via natural language and demonstrates robustness to unseen colours and object counts.
* The ablation studies are thorough and informative.)

**Key weaknesses**
* The conceptual distinction between TextOCVP and prior models (especially MAGE-DINO style token-based conditioning) is not articulated clearly. My issue here is not necessarily about whether the contribution is incremental or novel but rather about specific insight as to where TextOCVP differs from its baselines and whether performance changes due to the text-to-slot encoder.
* The interpretability analysis (attention visualisations) is descriptive but not explanatory; the paper misses an opportunity to provide deeper insights into why slot-level conditioning helps. For instance, it makes sense that the attention mechanisms learns a correlation between coordinates and object position as coordinates vary across scenes while the other tokens remain fixed, essentially being a discrete bottleneck. All these phenomena require a bit more explaining in my opinion and would make the paper immensely more valuable.
* The method inherits common limitations of slot-based models (like disentangled representations), but these are not discussed.
* It is unclear which improvements stem from the object-centric representations versus stronger architectural components (deeper transformer, DINO encoder), since these factors are somewhat confounded.

**Additional Comments:**

The paper is overall strong: well presented, empirically validated, and relevant to TMLR’s audience. My main concern is not with correctness or technical adequacy but with conceptual clarity and depth of insight. Strengthening the discussion around why slot-level conditioning helps, and what mechanisms underlie the observed behaviours, would elevate the contribution from a well-executed system paper to a more enlightening scientific result.

**Audience:**

Yes

**Audience Explanation:**

The intersection of object-centric learning, video prediction, and language grounding is highly relevant to the TMLR community, particularly for researchers working on goal-conditioned RL, robotics, and structured representations. The method is practical, addresses a real limitation of prior models (lack of controllable prediction), and introduces a structured alternative to recent diffusion-based approaches. Additionally, the CLIPort evaluation and focus on controllability make the work valuable for robotics and model-based decision-making tasks.

**Broader Impact Concerns:**

I do not see major ethical risks beyond those inherent to generative video models.

**Claims And Evidence:**

Yes

**Claims Explanation:**

The empirical claims regarding prediction quality, controllability, and robustness are supported by extensive experiments on two relevant datasets. The evaluation setup is well explained, ablations are thoughtful, and comparisons to strong baselines (MAGE, SEER, MAGEDINO, OCVP) are fair and comprehensive.

The qualitative results are meaningful and correspond to the claims.
However, while the experimental evidence convincingly demonstrates performance gains, some conceptual claims, such as the interpretability of the text-to-slot alignment or the inherent generalisation benefits of object-centric representations are stated but not deeply substantiated. The paper could be strengthened by providing more mechanistic insight into how and why the slot representations align with text attributes beyond the attention heatmaps.

Overall, the empirical evidence is strong, but the conceptual justification could benefit from deeper analysis.

**Requested Changes:**

1. Clarify the conceptual distinction between TextOCVP and MAGE/MAGEDINO-style models.
The paper currently frames the contribution as object-centric vs. holistic, but does not clearly articulate why slot-level text conditioning is fundamentally advantageous. A more explicit conceptual argument would greatly strengthen the contribution.

2. Provide deeper insight into the mechanisms behind text-to-slot alignment.
The attention visualizations are interesting but remain descriptive. The authors should include either:
a hypothesis grounded in representational learning (e.g., lower entropy slot space, conditional independence structure), or
an empirical study probing the specialization of slots or attention heads.
Without this, the interpretability claims are underspecified.

3. Discuss limitations of slot-based models.
The method inherits known issues from slot-based models. A short discussion acknowledging failure modes would make the work more balanced and informative. The authors touched on this in their results discussion (blurriness, lack of fine-grained detail), but like other parts of the paper they feel barely addressed. The authors do not necessarily have to run additional experiments for this but a deeper interpretation would help the reader take away more than from reading a simple technical report

---

> ### Author Response · Authors · 2025-11-28
> **Author response to Reviewer  AP1H**
>
> We thank the reviewer for their valuable feedback. Below we address all concerns and describe the changes made in the revised manuscript.
>
> ---
>
> **Regarding distinction between TextOCVP and MAGE/MAGE-DINO models**
>
> We appreciate the reviewer’s request for a clearer conceptual comparison.
> In the revision, we significantly expanded the comparative discussion of TextOCVP vs. MAGE/MAGE-DINO in Appendix C.1.
>
> In summary, we clarify that the key differences are two-fold:
>  - **Scene Representation**: TextOCVP uses object-centric slots while MAGE-style models rely on holistic VQ-token grids
>  - **Text Conditioning**:, MAGE uses a single global Motion Anchor, whereas TextOCVP applies text-to-slot cross-attention at every transformer layer, enabling object-wise and depth-wise integration of relevant textual cues.
>
> These clarifications are now discussed in detail in the updated Appendix C.1.
>
> ---
>
> **Deeper insight into mechanisms behind text-to-slot alignment**
>
> Thank you for raising this important conceptual point. While we did not perform a mechanistic interpretability study, we can offer additional high-level intuition for why slot-level conditioning supports alignment between text and objects.
>
> Object slots provide a compact, structured, and semantically meaningful latent space, where each slot encodes a representation of a single object (including attributes such as color, size, position, and shape). Because the latent space is lower-dimensional (with the number of objects being much smaller than the number of patches) and factorized across objects, it is naturally easier for text representations to attend to the correct slot than in holistic, patch-token models, where information about an object is dispersed across many spatial tokens, with unclear boundaries and no explicit grouping, making cross-modal alignment considerably harder.
>
> In practice, we observe that the text-to-slot attention maps reliably ground linguistic attributes (e.g., “blue block”, “small cube”) to the corresponding slot. Although this does not constitute a mechanistic explanation, it suggests that the structured nature of slots makes them more compliant to grounding than unstructured feature grids.
>
> A deeper mechanistic or empirical analysis—for instance, examining the geometry of the slot space, the emergence of slot specialization, or the roles of specific attention heads—would certainly be valuable. We agree that such work could provide richer insight into how and why the alignment emerges, and we view this as an exciting direction for future research.
>
> ---
>
> **Discussion of limitations of slot-based models**
>
> We expanded the Limitations section in Appendix A.1 to more explicitly discuss the existing limitations of slot-based object-centric models. Namely, we discuss the difficulties of such models to represent textures, small details, and tightly interlocking or highly deformable objects. Furthermore, we also discuss more fundamental limitations of object-centric models, such as the need for multi-resolution representations to parse scenes into part-whole hierarchical representations.
>
> ---
>
> **Regarding the source of improvements**
>
> To ensure a fair assessment of where the performance gains arise, our comparisons are designed so that architectural components (e.g., transformer depth, DINO encoder) do not confound the effect of the object-centric structure. More specifically:
>  - TextOCVP, Non-OC, and MAGE-DINO use the same encoder backbone and the same image decoder on CLIPort, thus ensuring a fair comparison. On CATER, MAGE even uses more expressive convolutional encoder and decoder modules  than TextOCVP and Non-OC.
>  - Non-OC uses the same predictor and backbone as TextOCVP, differing only in that it replaces object slots with a single holistic latent.
>
> Thus, the only major substantive difference is the latent scene representation, with Non-OC using a single, high-dimensional holistic latent vector per frame, MAGE/MAGE-DINO representing the scene with a grid of quantized latent vectors, and TextOCVP representing each observation with a set of object-centric slots.
>
> As shown in Tables 1 and 3, TextOCVP, with its slot-based latent space, outperforms all baselines for video prediction, as well as showing superior robustness and generalization properties. These results thus suggest that the object-centric design of TextOCVP’s latent space is the main source of such improved performance.

---

> > ### Comment · Reviewer_AP1H · 2025-12-22
> > **Final Assessment**
> >
> > I would like to thank the authors for the changes made in response to the rebuttal.
> >
> > Overall, I am happy with the amendments. In particular, I appreciate the added text-to-slot interpretation explaining why encoding text at the slot level can be more economical than conditioning on image patch representations. I think these kinds of conceptual insights meaningfully enrich the paper.
> >
> > I also accept the authors’ clarifications regarding the source of the observed improvements. The explanation that a structured, object-centric latent space helps ground text-to-image understanding is convincing. The additional clarification specifying which components are shared across methods and which differ makes it easier to assess the significance of the results. I would encourage the authors to make this distinction especially prominent in the main text, as it is central to understanding the contribution.
> >
> > Overall, I believe this is a good paper that represents a solid step toward fusing object-centric latent representations with text. While the experimental setup remains somewhat rudimentary, for instance, it is still unclear how the approach would scale to more complex visual scenes, I see this as a natural direction for future work rather than a critical limitation at this stage.
> >
> > I therefore recommend acceptance of this paper with the minor revisions performed by the authors.

---

### Review · Reviewer_PJwh · 2025-11-24

**Summary Of Contributions:**

This paper presents TextOCVP, a new object-centric approach for video prediction guided by textual descriptions. Past work tends to struggle to generalize and scale beyond simple scenes and don't afford easy ways to integrate additional information like language which has become a more popular input modality for robotics (e.g. VLAs). A text-to-slot attention mechanism is leveraged to allow text to attend to the image/video data for textual guidance. TextOCVP is tested on two benchmarks and outperforms the baselines.

Strengths:
- Paper is well written and articulated
- The presented ideas/tricks are great (such as text-to-slow attention) and would be particularly valuable to the CV and robotics community


Weaknesses:
- There aren't many ablations on TextOCVP itself and more of comparisons quantitatively and qualitatively with some previous baselines. This makes it difficult to determine the contributions of each proposed method and whether they do make a significant difference or not.
- Only synthetic datasets are tested, there are real world robotics video datasets that could be tested.

**Additional Comments:**

N/A

**Audience:**

Yes

**Audience Explanation:**

The paper tackles a highly relevant problem of world modeling with applications to robotics. The demonstration of the ability to integrate more multi-modal inputs makes this paper more interesting as robotics data tends to be highly multi-modal and will only grow in diversity and complexity in the future. For those who work more on the computer vision side of things, the approach to integrating text would be of interest to others in that field as well.

**Claims And Evidence:**

Yes

**Claims Explanation:**

Appropriate benchmarking is provided to compare TextOCVP against baselines with TextOCVP outperforming in terms of SSIM, LPIPS, and JEDi. Appropriate figures are provided to demonstrate the interpretability and controllability aspects of TextOCVP. Attention maps are visualized along with the input text/image sequences and show this well. They also show good results in terms of how object-centric features improve TextOCVP over the MAGE baseline.

**Requested Changes:**

While the results shown are great, it would be appreciated to see a proper ablation on whether the object-centric loss/part of the model is really necessary. The example shown in Figure 4a highlights it well, but is just a single example.

Moreover, only synthetic datasets are tested, however there are plenty of real world datasets in e.g. the OpenX Embodiment dataset work that could be compared against which can help better ground this paper.

---

> ### Author Response · Authors · 2025-11-28
> **Author response to Reviewer PJwh**
>
> We thank the reviewer for their valuable feedback. Below we address all concerns and describe the changes made in the revised manuscript.
>
> ---
>
> **Ablation of object-centricity**
>
> There might be a slight misunderstanding regarding the lack of ablation of object-centricity. The paper already includes comparisons with the Non-OC and MAGE/MAGE-DINO baselines, which help us directly ablate the effect of the latent representation. To make it clearer, we have added further details about the MAGE/MAGE-DINO and Non-OC baselines and their differences with respect to TextOCVP in Appendix C.1 and C.3, respectively.
>
> Specifically, Non-OC uses the same architecture as TextOCVP but replaces the slot-based latent space with a single, high-dimensional holistic latent embedding; whereas MAGE and MAGE-DINO use a similar encoder, predictor and decoder as TextOCVP, but represent the scene with a grid of latent vectors representing image patches. Therefore, the comparison with these baselines (especially with Non-OC) directly isolates the effect of the object-centric structure.
>
> Table 1 shows that TextOCVP consistently outperforms Non-OC and MAGE on both CATER and CLIPort, demonstrating the benefits of structured object-centric latent spaces for video prediction and dynamics modeling.
>
> Additionally, Figures 9, 13, 14, 15 and 16, as well as animations provided in the supplementary material, also qualitatively compare TextOCVP with the strong non-object-centric MAGE/MAGE-DINO baselines, and show the superior performance of our proposed model.
>
> Overall, these qualitative and quantitative comparisons demonstrate the benefits of object-centric latent spaces, in contrast to holistic, patch-based representations.
>
> ---
>
> **Lack of Ablations on TextOCVP model**
>
> The paper already includes several ablations and experiments that directly evaluate components of the TextOCVP architecture. Namely, Table 2 examines the effect of predictor depth, the use of residual connections, the number of slots, and the choice of text encoder. Furthermore, Appendix F.1 provides qualitative comparisons of different visual encoders (e.g., SAVi vs. DINOSAUR), illustrating how backbone selection impacts object representations and video rendering. Finally, Appendix F.8 introduces a new experiment comparing our original rendering module with a more expressive VQGAN-inspired decoder.
>
> Overall, these ablations provide a comprehensive understanding of how different architectural choices influence the behavior and performance of TextOCVP.
>
>
> ---
>
>
> **Regarding the application to real-world robotics datasets**
>
> We appreciate the reviewer's interest in extending our work to real-world settings. Our current work was intended as an initial exploration into language-conditioned object-centric video prediction, emphasizing controlled, robust, and interpretable evaluations across synthetic but diverse and well-structured environments (CATER,  CLIPort). These datasets allow us to rigorously analyze controllability and object grounding—key aspects of TextOCVP.
>
> We agree that real-world generalization is a necessary next step. Our framework is modular and scalable, and we envision future extensions that integrate powerful vision-language encoders and more expressive rendering modules. We discuss such extensions in Appendix A, and plan to pursue them in future work.

---

### Author Response · Authors · 2025-11-28
**General response from authors**

We sincerely thank all reviewers for their thorough, thoughtful, and constructive feedback. We are encouraged that all reviewers find our core idea compelling and well motivated, our empirical claims to be well supported, our results convincing and clearly presented, and our contributions relevant to the robotics, computer vision, and representation learning communities.

We also appreciate the positive remarks from the reviewers regarding the clarity of the paper (PJwh, AP1H), the practical value of text-conditioned slot representations (PJwh, AP1H, q6ZE), the strength of our empirical evaluation, including robustness, generalization and controllability (AP1H, q6ZE), our thorough and informative ablation studies (AP1H), and that overall our paper is strong (AP1H).

We have revised our manuscript based on your feedback and provide a list of changes below. We have also responded to your individual comments.

**Summary of revisions**: We summarize below the changes made to our manuscript; these changes have also been highlighted in green in the new version:
 - **Softening claims related to robotics**, ensuring the terminology accurately reflects evaluation in simulated robotic environments.
 - **A discussion on inherent limitations of slot-based models** (Appendix A.1), including challenges with fine detail, deformable and articulated objects, and the absence of multi-resolution structure.
 - **A clearer comparison between MAGE/MAGE-DINO and TextOCVP** (Appendix C.1), emphasizing representational and conditioning differences.
 - **Additional details on the Non-OC baseline** (Appendix C.3), and how it helps us ablate the effect of object-centric latent spaces.
 - **An experiment using a more expressive VQGAN-inspired decoder** (Appendix F.8) to investigate the importance and effect of the video rendering module.
 - **An evaluation of prediction quality over long horizons** (Appendix F.9), including plots of LPIPS and Slot-Mask-IoU over time.

We thank the reviewers again for their constructive feedback. We believe that all comments have been addressed in this revision, but we will be happy to address any further comments from reviewers.

Best regards,

Authors of TextOCVP

---

### Decision · Action_Editor_TD6z · 2026-01-02

**Recommendation:** Accept as is

**Audience:**

Yes

**Audience Explanation:**

The paper represents a solid step toward fusing object-centric latent representations with text.

**Claims And Evidence:**

Yes

**Claims Explanation:**

The explanation that a structured, object-centric latent space helps ground text-to-image understanding is convincing.